# Few-Shot Knowledge Distillation of LLMs With Counterfactual Explanations

**Faisal Hamman**    **Pasan Dissanayake**    **Yanjun Fu**    **Sanghamitra Dutta**
University of Maryland, College Park
{fhamman, pasand, yanjunfu, sanghamd}@umd.edu

## Abstract

Knowledge distillation is a promising approach to transfer capabilities from complex teacher models to smaller, resource-efficient student models that can be deployed easily, particularly in task-aware scenarios. However, existing methods of task-aware distillation typically require substantial quantities of data which may be unavailable or expensive to obtain in many practical scenarios. In this paper, we address this challenge by introducing a novel strategy called **Co**unterfactual-explanation-infused **D**istillation (CoD) for *few-shot task-aware knowledge distillation by systematically infusing counterfactual explanations*. Counterfactual explanations (CFEs) refer to inputs that can flip the output prediction of the teacher model with minimum perturbation. Our strategy CoD leverages these CFEs to precisely map the teacher's decision boundary with significantly fewer samples. We provide theoretical guarantees for motivating the role of CFEs in distillation, from both statistical and geometric perspectives. We mathematically show that CFEs can improve parameter estimation by providing more informative examples near the teacher's decision boundary. We also derive geometric insights on how CFEs effectively act as knowledge probes, helping the students mimic the teacher's decision boundaries more effectively than standard data. We perform experiments across various datasets and LLMs to show that CoD outperforms standard distillation approaches in few-shot regimes (as low as 8 - 512 samples). Notably, CoD only uses half of the original samples used by the baselines, paired with their corresponding CFEs and still improves performance. Our code is available at https://github.com/FaisalHamman/CoD.

## 1 Introduction

Large Language Models (LLMs) have demonstrated state-of-the-art performance across a broad spectrum of tasks [1–3]. However, as the size of LLMs grow, so does the associated computational burden, making them difficult to deploy in resource-constrained environments, e.g., mobile phones, edge devices, and embedded systems [4]. The challenge, therefore, lies in making large models more efficient and accessible without sacrificing performance. To this end, knowledge distillation (KD) (initially proposed in [5]; see surveys [6–8]) has emerged as a powerful technique for model compression, enabling smaller student models to mimic the performance of a larger teacher model. In the context of LLMs, KD plays a central role in transferring the broad capabilities such as natural language understanding [9], reasoning [10], instruction following [11] onto smaller models.

While LLMs are trained for a broad range of tasks, we may often want a smaller, task-specific language model when full task coverage is not required, particularly in resource-constrained environments. To support this, *task-aware knowledge distillation* [12, 13] has been proposed to selectively transfer task-relevant knowledge from teacher to student models. While effective, these methods typically assume access to large datasets [14]. However, in many real-world applications, the amount of data

39th Conference on Neural Information Processing Systems (NeurIPS 2025).

available is often limited [15–18], and obtaining high-quality human-annotated data is expensive. Despite advances on algorithmic strategies for task-aware KD in LLMs [14], the problem of data selection for KD has received limited interest, particularly in few-shot settings. In this work, we study *few-shot and task-aware knowledge distillation for LLMs*, where student models are distilled from teacher models using a very small number of samples labeled for a task (also called shots). Few-shot task-aware distillation remains underexplored for LLMs. In classical ML, few-shot training has poor generalization [19], and thus causes ineffective distillation due to insufficient task coverage [20, 21]. However, few-shot distillation holds potential for LLMs because they are pretrained on a large corpora, also drawing inspiration from the prior success of few-shot learning [22].

In this work, we propose a few-shot task-aware knowledge distillation strategy by systematically integrating a type of posthoc explainability technique called counterfactual explanations (CFEs) [23, 24]. CFEs are inputs that can flip the output prediction of a model with minimum perturbations. We find that CFEs can act as knowledge probes, helping the students mimic the teacher's decision boundaries more effectively than standard data. Our work bridges explainability and model compression by turning explanations into actionable training signals, guiding the student into learning the teacher's decision-making process more effectively. This results in more faithful knowledge transfer even with very limited data. Our contributions can be summarized as follows:

- **A counterfactual explanation-based strategy for few-shot distillation.** We propose a novel framework CoD, short for **Co**unterfactual-explanation-infused **D**istillation, for task-aware knowledge distillation under few-shot regimes. By enriching the few-shot training set with CFEs, we improve the student's ability to mimic the fine-grained details of the teacher's decision boundary with fewer labeled examples. We validate this intuition through a synthetic experiment on the 2D `moons` dataset, showing that CFE-infused distillation better replicates the teacher's decision surface compared to using standard few-shot samples (see Fig. 1 & Fig. 3).

- **Theoretical guarantees motivating the role of CFEs in distillation.** We provide theoretical guarantees that serve as motivation for our approach, from both statistical and geometric perspectives. First, in a logistic regression setting, we show that CFEs improve parameter estimation by maximizing the Fisher Information (see Def. 2 & Thm. 1). Our proof specifically leverages the fact that the CFEs lie quite close to the decision boundary to show that they reduce the expected estimation error of the student model compared to standard distillation. Next, moving beyond statistical guarantees and linear models, we also provide a geometric analysis for non-linear models, establishing that if a student matches the teacher's predictions on the original data and their counterfactual pairs, then their decision boundaries will remain close: this is quantified by a provably small *Hausdorff distance*, a formal measure of distance between two subsets within a space (see Def. 3 & Thm. 2).

- **Empirical validation.** We evaluate CoD on six benchmark datasets using `DeBERTa-v3` [25] and `Qwen2.5` [2] model families. We compare against strong baselines including standard Knowledge Distillation (KD) [5], Layer-wise Distillation (LWD) [13], and Task-aware layer-wise Distillation (TED) [12] under various few-shot settings ($k = 8$, 16, 32, 64, 128, and 512). Our results demonstrate that CoD consistently outperforms baselines in few-shot regimes, with significant improvements in extremely data-scarce scenarios ($k \leq 64$). Notably, CoD only uses *half of the original labeled samples used by the baselines* (i.e., $k/2$ original infused with their corresponding $k/2$ CFEs, leading to $k$ shots), and still gives improved performance. For instance, with $k = 8$ samples on `IMDB` dataset, LWD + CoD improves over standard LWD by more than 10 points.

**Related Works:** Knowledge distillation has emerged as a powerful framework for model compression [5]. While early works focused on transferring soft labels via output logits [26], subsequent advances explored richer supervision signals such as intermediate feature alignment [13, 27–29]. As LLMs grow in size and inference cost [30, 31], distillation has become increasingly important for transferring capabilities into smaller models [14, 6–8]. More recently, task-aware knowledge distillation for LLMs has gained traction, aiming to selectively distill knowledge relevant to a specific downstream task [12, 32]. Despite these algorithmic innovations [12], there has been relatively little focus on data selection for distillation, particularly in few-shot settings. Most prior works assume ample training data, leaving few-shot knowledge distillation largely underexplored. While some works [20, 16–18] have studied distillation in classical ML under low-data regimes, they do not address the challenges specific to distilling LLMs. In this work, we establish the paradigm of few-shot distillation in LLMs by integrating explainable data selection. Our work is broadly aligned with the spirit of data-efficient ML, which aims to improve performance under limited supervision [33–36].

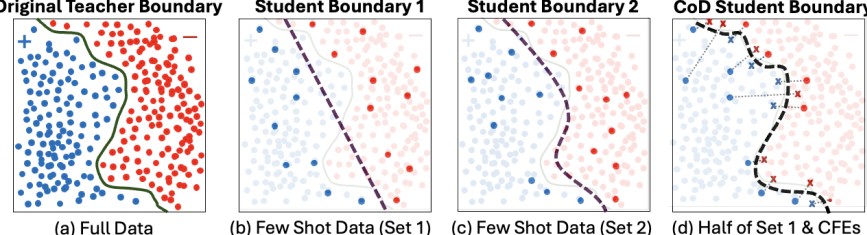

| Original Teacher Boundary | Student Boundary 1 | Student Boundary 2 | CoD Student Boundary |
|---|---|---|---|
| (a) Full Data | (b) Few Shot Data (Set 1) | (c) Few Shot Data (Set 2) | (d) Half of Set 1 & CFEs |

Figure 1: **Intuition behind our approach**: *(a)* Teacher trained on the full dataset with true decision boundary. *(b–c)* With few-shot supervision, many classifiers can fit the sparse points; the resulting student boundaries (dashed lines) can vary and do not always align with the teacher's boundary (unfaithful distillation). *(d)* Pairing each point with its CFE (×, linked to originals) during distillation makes the student match the teacher's soft predictions at these points. CFEs act as boundary-near pegs that clamp the student to the teacher's decision surface, producing a more faithful distillation even under few-shot budgets.

Counterfactual explanations (CFEs) [23, 24, 37–43] have been widely studied in classical ML, particularly for algorithmic recourse in high-stakes applications such as finance, healthcare, and law. An interesting work [44] uses CFEs for model reconstruction by deriving theoretical relationships between reconstruction error and the number of counterfactual queries using polytope theory. In the natural language domain, some methods have been proposed to generate semantically valid CFEs using either token-level perturbations [45] or controlled generation with language models [46, 47, 45], but they have not been integrated for knowledge distillation. Another line of work is counterfactual reasoning in causal inference [48], where the goal is to estimate the effect of interventions under a structural causal model, which is different from our objectives. Counterfactual data have been used to address the issue of spurious patterns [49–51], improve generalization [52, 53], and enhance performance on out-of-distribution data [54, 55]. In contrast, our work studies the role of CFE in few-shot task-aware distillation, specifically aligning the student with the teacher's outputs to mimic the teacher's decision boundary more effectively.

## 2  Preliminaries

LLMs are highly effective for natural language processing. Built upon the transformer architecture [56], LLMs consist of multiple stacked layers, each containing a multi-head self-attention mechanism followed by a position-wise feed-forward neural network. Let $g(\cdot; \theta)$ denote a transformer-based model parameterized by $\theta$. The model takes an input sequence $\mathbf{x} \in \mathcal{X}$ where $\mathcal{X}$ is the input space. The model output is a probability distribution over the vocabulary space, but for task-aware settings such as sentiment analysis, it is a probability distribution over $C$ class labels, i.e., $g : \mathcal{X} \rightarrow [0,1]^C$. The loss function is defined as: $\mathcal{L}(\theta) = \mathbb{E}_{\mathbf{x} \sim \mathcal{X}}[\ell(g(\mathbf{x}; \theta))]$, where $\ell$ denotes the task-specific loss, such as cross-entropy for classification tasks or causal language modeling loss for generative models.

**Knowledge Distillation (KD).** KD is a technique that transfers knowledge from a large, pre-trained teacher model to a smaller, student model [57]. Let $g_t(\cdot; \theta_t)$ be the teacher model with parameters $\theta_t$ and $g_s(\cdot; \theta_s)$ be the student model with parameters $\theta_s$. The teacher model $g_t(\cdot; \theta_t)$ provides soft labels to assist in training the student model $g_s(\cdot; \theta_s)$. The student is trained using a loss function that is a combination of the task-specific loss and the distillation loss as follows: $\min_{\theta_s} \mathcal{L}(\theta_s) + \alpha \mathcal{L}_{\text{KD}}(\theta_t, \theta_s)$. Here, $\mathcal{L}(\theta_s)$ is the task-specific loss, e.g., the cross-entropy loss between the student's outputs and true-labels, and $\mathcal{L}_{\text{KD}}(\theta_t, \theta_s) = \mathbb{E}_{\mathbf{x} \sim \mathcal{X}}[d(g_t(\mathbf{x}; \theta_t), g_s(\mathbf{x}; \theta_s))]$ is the distillation loss which captures the distance between the outputs of the teacher and student. Typically, the distance is computed using the Kullback-Leibler (KL) divergence, i.e., $\text{KL}(g_t(\mathbf{x}; \theta_t) \| g_s(\mathbf{x}; \theta_s)) = \sum_{c=1}^{C} g_t^{(c)}(\mathbf{x}; \theta_t) \log \frac{g_t^{(c)}(\mathbf{x}; \theta_t)}{g_s^{(c)}(\mathbf{x}; \theta_s)}$, where the superscript $(c)$ is for the assigned probability for class $c$ by each model.

**Layer-Wise Distillation (LWD).** In large transformer-based models, the teacher's outputs may not fully capture the knowledge embedded in intermediate layers. Beyond matching final outputs, one can also align the intermediate features of the teacher and student [13]. At a few selected layers, the teacher's hidden activations $h_t^l$ and the student's activations $h_s^l$ (optionally projected into the same dimension) are computed and their difference is also penalized using a mean-squared-error loss [13].

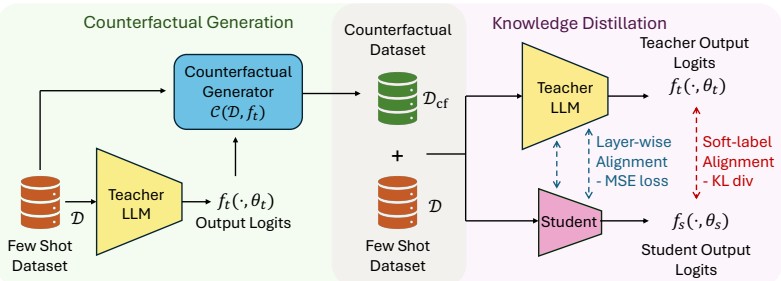

Figure 2: Overview of our framework: **Co**unterfactual Explanation-Infused **D**istillation (CoD).

The student is trained using a loss as follows:

$$\min_{\theta_s} \quad \mathcal{L}(\theta_s) \ + \ \alpha \, \mathcal{L}_{\text{KD}}(\theta_t, \theta_s) \ + \ \beta \, \mathcal{L}_{\text{LWD}}(\theta_t, \theta_s) \tag{1}$$

Here, $\mathcal{L}_{\text{LWD}}(\theta_t, \theta_s)$ is the additional layer-wise alignment term added alongside the task-specific loss and distillation loss, e.g., $\mathbb{E}_{\mathbf{x} \sim \mathcal{X}}[\sum_{l \in \mathcal{I}} \| h_t^l - h_s^l \|_2^2]$ where $\{h_t^l, h_s^l\}_{l \in \mathcal{I}}$ are the teacher and student activations for a given input $\mathbf{x}$ over a set $\mathcal{I}$ of layers, and $\alpha, \beta \geq 0$ balance the three objectives.

**Counterfactual Explanations (CFEs).** Given a model's decision on an input $\mathbf{x}$, a CFE [23, 24] finds the minimal modification $\mathbf{x}'$ such that the model's output changes in a desired way. These explanations help interpret model decisions and provide actionable guidance to users to flip the prediction. In our context, we look into CFEs in the NLP domain where the inputs are token sequences. A counterfactual in this setting is a minimally perturbed sentence that causes the teacher LLM's prediction to flip. For instance, given the sentence *I loved the movie*, labeled as positive sentiment, a CFE would be *I hated the movie*, a semantically similar but sentiment-flipped variant.

**Our Problem Setting.** We consider a binary classification setting where the teacher model will be denoted as $f_t : \mathcal{X} \rightarrow [0, 1]$. The input space $\mathcal{X} \subseteq \mathbb{R}^{n \times d}$, with $n$ being the sequence length and $d$ is the model dimension, after the entire input sequence has already been passed through the tokenizer and embedding layers of the LLM. The teacher model $f_t(\mathbf{x})$ gives the class-1 probability output of the model for input $\mathbf{x}$, i.e., $f_t(\mathbf{x}) := g_t^{(1)}(\mathbf{x}; \theta_t)$, where the superscript (1) is for the assigned probability for class 1. The final predicted class is given by $\hat{f}_t(\mathbf{x}) = \mathbb{I}\,[f_t(\mathbf{x}) \geq 0.5] \in \{0, 1\}$.

**Definition 1** (Closest CFE $\mathcal{C}(\mathbf{x}, f_t)$). *Given $\mathbf{x} \in \mathbb{R}^{n \times d}$ such that $f_t(\mathbf{x}) < 0.5$, the closest CFE is a point $\mathbf{x}' \in \mathbb{R}^{n \times d}$ with opposite prediction that minimizes the Frobenius-norm $\|\mathbf{x} - \mathbf{x}'\|_F$:*

$$\mathcal{C}(\mathbf{x}, f_t) = \underset{\mathbf{x}' \in \mathbb{R}^{n \times d}}{\arg \min} \|\mathbf{x} - \mathbf{x}'\|_F \text{ such that } f_t(\mathbf{x}') \geq 0.5. \tag{2}$$

Definition 1 naturally extends to multiclass settings, where a CFE can be defined as the minimum perturbation that changes the predicted class to any other target class.

**Remark 1** (Data Manifold Counterfactual Explanations). *In practice, unconstrained counterfactuals may lead to unrealistic or out-of-distribution examples. To address this, we can constrain $\mathbf{x}'$ to lie within the data manifold $\mathcal{X}' \subseteq \mathbb{R}^{n \times d}$, ensuring that generated counterfactuals remain semantically plausible. These data-manifold counterfactuals preserve natural language structure. In our work, we use a hybrid generation strategy that combines LLM-based prompting with teacher model feedback to generate such data-manifold CFEs. Further details are provided later in Section 3.*

Given a training data budget $k$ (few-shots) and a teacher model $f_t$, our **goal** is to distill a smaller student model $f_s : \mathcal{X} \rightarrow [0, 1]$ with high-performance at a specific task by leveraging CFEs.

## 3 Main Contributions

We begin with an experiment on 2D synthetic data that demonstrates how CFEs help student models mimic the teacher's decision boundary more effectively than standard data. Next, we provide theoretical results motivating our approach from both statistical and geometric perspectives. Finally,

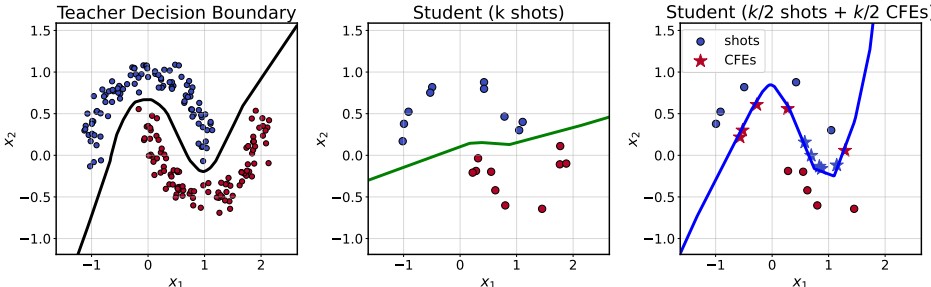

Figure 3: **Decision boundaries for teacher and two students trained on synthetic data under few-shot setting.** The teacher *(a)* is trained on the full dataset and serves as the distillation target. Student *(b)* is distilled using 20 randomly sampled data points, and results in a poorly aligned decision boundary with the teacher. Student *(c)* is also trained on 20 total samples, 10 original data points and their 10 CFEs. This student learns a better decision boundary that aligns more closely with the teacher. This is because the CFEs lie close to the teacher's decision boundary and the KD loss encourages the student to match the teacher's soft predictions at the CFEs, clamping the student's boundary to the teacher's boundary.

we describe our CFE generation pipeline for natural language inputs, which leverages LLMs to produce semantically plausible CFEs, leading to our proposed framework CoD.

**Synthetic Dataset Experiments to Illustrate the Role of CFE in Distillation:** We conduct experiments on the 2D `moons` dataset [58] and show that infusing few-shot data with CFEs significantly improves student-teacher alignment in distillation (see Figure 3). We train a *teacher* model—a two-layer neural network with architecture $[2 \to 64 \to 64 \to 2]$ on the full dataset. The *student* network has a smaller architecture $[2 \to 16 \to 2]$. We randomly sample $k = 20$ original points (10 per class). For the original points, we compute their closest CFE, a minimally perturbed input that flips the teacher's predicted class. We follow a gradient-based method [23] to compute CFEs by perturbing each point in the direction of the teacher's logit margin until the predicted class flips. We consider two student models: one trained on the $k$ few-shot samples alone, and another trained on $k/2$ few-shot samples and their CFEs. In both cases, we perform knowledge distillation by minimizing a combination of cross-entropy loss on the hard labels and KL-divergence between the student and teacher soft predictions. Figure 3 shows the decision boundaries of the teacher, the baseline student, and the CFE-infused student. CFEs cluster near the decision boundary, enriching the distillation data in high-uncertainty regions. The student trained with CFEs aligns more closely with the teacher, thus motivating the use of boundary-targeted examples for improved knowledge distillation.

**Statistical Guarantees Motivating Our Approach:** Here, we provide a theoretical motivation for the use of CFEs in few-shot knowledge distillation. We analyze a logistic regression setting using a measure from estimation theory called Fisher Information [59] (also see Definition 2) that captures the information contained by a random variable about a parameter to be estimated. We show that a dataset containing CFEs, which essentially lie much closer to the teacher's decision boundary, yields a Fisher Information Matrix with higher overall information content for parameter estimation. As a result, the student's expected estimation error is lower compared to training on standard samples.

**Definition 2** (Fisher Information Matrix [59]). *Let $\mathcal{L}(\theta)$ be the log-likelihood of a parametric distribution $p(y, x; \theta)$, where $\theta$ is the parameter vector to be estimated. The* Fisher Information Matrix *(FIM) at parameter $\theta$ is defined as:*

$$\mathcal{I}(\theta) = \mathbb{E}_{\mathbf{x},y} \left[ \nabla_\theta \log p(y, \mathbf{x}; \theta) \nabla_\theta \log p(y, \mathbf{x}; \theta)^\top \right].$$

Intuitively, Fisher Information measures the curvature of the log-likelihood: flatter regions (low curvature) imply high uncertainty in estimating $\theta$, while sharper regions (high curvature) indicate that small changes in $\theta$ cause large changes in likelihood, enabling more precise parameter estimation.

We consider a binary classification setting where both the teacher and student are logistic regression models. Suppose the teacher, parameterized by $\mathbf{w}_t$, defines the true data-generating distribution with predicted probabilities $p_t(y = 1|\mathbf{x}) = \sigma(\mathbf{w}_t^\top \mathbf{x})$ where $\sigma(\cdot)$ is the softmax function. Suppose, the student, with parameters $\mathbf{w}_s$, is obtained via maximum likelihood estimation (MLE) [59] using either a standard dataset $\mathcal{D}$ or a CFE-infused dataset $\mathcal{D}_{\text{cf}}$. Since the CFEs lie close to the teacher's decision boundary, we have $\mathbf{w}_t^\top \mathbf{x}_c \approx 0$ when $\mathbf{x}_c$ is a CFE.

**Theorem 1** (CFEs Improve Model Parameter Estimation). *Let $\mathbf{w}_s$ and $\mathbf{w}_s^{(\text{cf})}$ be the student parameters obtained via MLE on $\mathcal{D}$ (standard) and $\mathcal{D}_{\text{cf}}$ (CFE-infused). Assuming the teacher's parameters $\mathbf{w}_t$ capture the true data-generating distribution, that CFEs lie near the decision boundary, and that the second moments $\mathbb{E}_{\mathbf{x}}[\mathbf{x}\mathbf{x}^\top] \approx \mathbb{E}_{\mathbf{x}_c}[\mathbf{x}_c \mathbf{x}_c^\top]$. Then estimation error satisfies:*

$$\mathbb{E}\left[\|\mathbf{w}_s^{(\text{cf})} - \mathbf{w}_t\|^2\right] < \mathbb{E}\left[\|\mathbf{w}_s - \mathbf{w}_t\|^2\right].$$

*Proof Sketch:* The key step in our proof relies on showing that the Fisher Information is given by $\mathcal{I}(\mathbf{w}_t; \mathcal{D}) = \sum_i p_t(y = 1|\mathbf{x}_i)(1 - p_t(y = 1|\mathbf{x}_i))\mathbf{x}_i \mathbf{x}_i^\top$. The scalar weight $p_t(y = 1|\mathbf{x})(1 - p_t(y = 1|\mathbf{x}))$ is maximized when $p_t(y = 1|\mathbf{x}) = 0.5$, i.e., $\mathbf{x}$ lies on the decision boundary. Standard samples in few-shot settings typically lie far from the boundary and contribute little to the FIM, whereas CFEs are constructed to lie near it and thus contribute significantly more. As a result, the FIM of the CFE-infused dataset $\mathcal{D}_{\text{cf}}$ dominates that of the standard dataset $\mathcal{D}$ in Loewner order [60] (i.e., $\mathcal{I}(\mathbf{w}_t; \mathcal{D}_{cf}) \succ \mathcal{I}(\mathbf{w}_t; \mathcal{D})$). The CFE-infused dataset provides strictly more information for parameter estimation than the standard dataset, ultimately leading to the bound on expected estimation error.

The full proof is in Appendix A. Notably, while this result mathematically motivates the advantages of CFEs in few-shot distillation, it still assumes linear models and same student-teacher capacity (size). For more general non-linear settings, we provide a geometric perspective as discussed next.

**Geometric Insight for Using CFEs for Distillation:** Here, we examine the geometric effect of CFEs on student-teacher alignment in non-linear settings. Specifically, we show that when data points and their CFE pairs are included during distillation, the student's decision boundary comes much closer to the teacher's boundary, as quantified by a formal measure called *Hausdorff distance* [61] between their respective decision surfaces. The Hausdorff distance (see Figure 4) captures the worst-case discrepancy between two sets (in our case, the decision boundaries of the teacher and student models) by quantifying how far any point on one boundary is from the closest point on the other.

Let $\mathcal{M}_t = \{\mathbf{x} \in \mathbb{R}^{n \times d} \mid f_t(\mathbf{x}) = 0.5\}$ and $\mathcal{M}_s = \{\mathbf{x} \in \mathbb{R}^{n \times d} \mid f_s(\mathbf{x}) = 0.5\}$ denote the decision boundaries of the teacher and the student. Our goal is to examine how close is the student's decision boundary to the teacher's. To quantify this alignment, we define the Hausdorff distance as follows:

**Definition 3** (Hausdorff Distance). *Let $\mathcal{M}_t, \mathcal{M}_s \subseteq \mathbb{R}^{n \times d}$ be two non-empty subsets of a metric space. The* Hausdorff distance *is defined as:*

Figure 4: Hausdorff Distance.

$$\mathrm{H}(\mathcal{M}_s, \mathcal{M}_t) = \max\left\{\sup_{\mathbf{x} \in \mathcal{M}_t} \inf_{\mathbf{u} \in \mathcal{M}_s} \|\mathbf{x} - \mathbf{u}\|_F, \ \sup_{\mathbf{u} \in \mathcal{M}_s} \inf_{\mathbf{x} \in \mathcal{M}_t} \|\mathbf{u} - \mathbf{x}\|_F\right\}.$$

We observe that for training sample $\mathbf{x}_i$ and its CFE $\mathbf{x}_i'$, the segment joining them cuts the teacher's boundary since they have different predictions. Essentially, there exists an intersection point $\mathbf{x}_i^\star$ on this segment such that $f_t(\mathbf{x}_i^\star) = 0.5$. Now, if the student is taught to matches the teacher at $\mathbf{x}_i$ and $\mathbf{x}_i'$, the student would also have another intersection point on this segment. These two intersection points lying on the teacher and student boundaries will act as clamps, pulling the two boundaries close to each other, since their own gap gets smaller as $\mathbf{x}_i$ and its CFE $\mathbf{x}_i'$ comes closer. We assume boundaries are closed and distance is measured within a compact region (e.g., support of data).

**Lemma 1** (Existence of Boundary Crossing for Counterfactual Pairs). *Let $f_t : \mathbb{R}^{n \times d} \to [0, 1]$ be a continuous function. For a datapoint and its counterfactual pair $(\mathbf{x}_i, \mathbf{x}_i')$, there exists a point $\mathbf{x}_i^\star = \alpha \mathbf{x}_i + (1 - \alpha)\mathbf{x}_i'$ for an $\alpha \in (0, 1)$ (on the line joining $\mathbf{x}_i$ and $\mathbf{x}_i'$) such that: $f_t(\mathbf{x}_i^\star) = 0.5$.*

**Theorem 2** (Teacher–Student Boundary Proximity). *Let $f_t$, $f_s : \mathbb{R}^{n \times d} \to; [0, 1]$ be the teacher and student model, with decision boundaries $\mathcal{M}_t = \{\mathbf{x} \mid f_t(\mathbf{x}) = 0.5\}$ and $\mathcal{M}_s = \{\mathbf{x} \mid f_s(\mathbf{x}) = 0.5\}$, respectively. Assume we observe a CFE-infused dataset $\mathcal{D}_{cf} = \left\{(\mathbf{x}_i, \mathbf{x}_i')\right\}_{i=1}^k$ satisfying: (A1) Minimal perturbation: $\|\mathbf{x}_i - \mathbf{x}_i'\|_F \leq \alpha$ with $\alpha > 0$; (A2) Exact distillation: $f_s(\mathbf{x}_i) = f_t(\mathbf{x}_i)$ and $f_s(\mathbf{x}_i') = f_t(\mathbf{x}_i')$; and (A3) $\varepsilon$-spread along the teacher and student boundary, i.e., for each pair, there exist a teacher's (or student's) crossing point $\mathbf{x}_i^\star = \alpha \mathbf{x}_i + (1 - \alpha)\mathbf{x}_i'$ for $\alpha \in (0, 1)$ such that $f_t(x_i^\star) = 0.5$ (or, $f_s(x_i^\star) = 0.5$) and for every $a \in \mathcal{M}_t$ (or $\mathcal{M}_s$), there exists an $i$ with $\|a - \mathbf{x}_i^\star\|_2 \leq \varepsilon$. Then the Hausdorff distance between the decision boundaries obeys: $\mathrm{H}(\mathcal{M}_s, \mathcal{M}_t) \leq \alpha + \varepsilon$.*

Consequently, tight (small $\alpha$) and well-spread (small $\varepsilon$) CFE pairs guarantee that the student boundary remains inside an $(\alpha + \varepsilon)$-tube around the teacher boundary.

**Interpretation of the assumptions and bound**. Our theorem makes three intuitive assumptions. *(A1)* Minimal perturbation requires each input and its CFE pair $(\mathbf{x}, \mathbf{x}')$ to differ by at most $\alpha$. CFEs are by definition the minimal changes that flips the teacher's prediction, so $\alpha$ is typically much smaller than the distance between arbitrary training points (note that we do no need CFEs to sit exactly on the teacher's boundary, i.e., $f_t = 0.5$). It suffices that the perturbation is small and flips the label—capturing the practical way CFEs are produced. *(A2)* Exact distillation agreement assumes the student matches the teacher's outputs on the input and CFE pairs. This is reasonable, as these examples are directly used in training, and their logits are aligned through the distillation (KL) loss. *(A3)* $\varepsilon$-spread assumes the inputs are reasonably well spread. No region of the teacher's or student's boundary is more than $\varepsilon$ away from a crossing point (generally smooth). Under these assumptions, the Hausdorff gap between student and teacher boundaries is tightly bounded by $\alpha + \varepsilon$. This ensures the student's decision boundary stays within an $(\alpha + \varepsilon)$-tube around the teacher's, illustrating the geometric faithfulness we want in few-shot knowledge distillation. See proofs in Appendix B.

**Proposed Algorithm (CoD).** We propose CoD, a **C**ounterfactual Explanation-infused **D**istillation strategy for few-shot, task-aware distillation of LLMs. The first step is CFE generation. Existing methods primarily fall into optimization-based [23], search-based [62], and generative approaches [63]. These methods can be computationally expensive for LLMs, and frequently yield out-of-distribution or semantically implausible examples. To address this, we adopt a hybrid approach that combines the teacher model predictions with an LLM as an oracle for CFE generation. Specifically, given an input and its original label, we prompt an LLM (e.g., GPT-4o [64]) to generate a semantically similar sentence intended to flip the label with minimal changes to the input. We then check whether this generated example indeed flips the teacher model's prediction, ensuring its utility as a true CFE. Once validated, each CFE is paired with its original input $(\mathbf{x}, \mathbf{x}')$ and added to the training set. During distillation, we ensure that each input–CFE pair is included in the same mini-batch, enabling the

---

**Algorithm 1** CoD: CFE-infused Distillation

**Require:** Teacher $g_t$, student $g_s$, dataset $\mathcal{D}_k = \{(\mathbf{x}_i, y_i)\}_{i=1}^k$, CFGen, learning rate $\eta$, loss weights $\alpha$ (KD), $\beta$ (LWD), Epochs $E$
1: $\mathcal{D}_{\text{cf}} \leftarrow \emptyset$
2: **for all** $(\mathbf{x}, y) \in \mathcal{D}_k$ **do**
3: $\quad x' \leftarrow \text{CFGen}(\mathbf{x}, g_t)$
4: $\quad \mathcal{D}_{\text{cf}} \leftarrow \mathcal{D}_{\text{cf}} \cup \{(\mathbf{x}', 1 - y)\}$
5: **end for**
6: $\mathcal{D}_{\text{train}} \leftarrow \mathcal{D}_k \cup \mathcal{D}_{\text{cf}}$
7: **for** $e = 1$ **to** $E$ **do**
8: $\quad$ **for all** $(\mathbf{x}, y) \in \mathcal{D}_{\text{train}}$ **do**
9: $\quad\quad \mathcal{L}_{\text{hard}} \leftarrow \text{CE}(g_s(\mathbf{x}), y)$
10: $\quad\quad \mathcal{L}_{\text{KD}} \leftarrow \text{KL}(g_t(\mathbf{x}) \| g_s(\mathbf{x}))$
11: $\quad\quad \mathcal{L}_{\text{LWD}} \leftarrow \sum_{l \in \mathcal{I}} \|h_t^{(l)} - h_s^{(l)}\|_2^2$
12: $\quad\quad \mathcal{L} \leftarrow \mathcal{L}_{\text{hard}} + \alpha\,\mathcal{L}_{\text{KD}} + \beta\,\mathcal{L}_{\text{LWD}}$
13: $\quad\quad$ Update $\theta_s \leftarrow \theta_s - \eta\,\nabla_{\theta_s}\mathcal{L}$
14: $\quad$ **end for**
15: **end for**
16: **return** distilled student $g_s$

---

student to jointly learn from both examples. The student is then trained using a combination of task loss, KL-based distillation loss, and optional layer-wise alignment. An overview of this process is described in Algorithm 1, with full implementation details and prompts provided in Appendix C.

# 4 Experiments

The goal of our experiments is to evaluate the effectiveness of integrating CFEs for knowledge distillation under few-shot learning settings. We investigate whether using limited real samples infused with their corresponding CFEs enable better distillation compared to only using real samples.

**Datasets.** We evaluate CoD across six text classification benchmarks that span a range of domains. SST2 is a binary sentiment classification task derived from movie review snippets [65]. Sentiment140 consists of tweets labeled as positive or negative, reflecting user sentiment in short social media posts [66]. IMDB is a binary sentiment classification dataset containing full-length movie reviews [67]. CoLA (Corpus of Linguistic Acceptability) is a grammaticality judgment task that requires the model to identify whether a sentence is linguistically acceptable [68]. Amazon Polarity contains customer reviews labeled as positive or negative sentiment [69]. Yelp is another sentiment classification dataset based on user-generated restaurant reviews [70].

**Model.** We experiment with two prominent model families: DeBERTa-v3 [25] and Qwen2.5 [2]. For DeBERTa-v3, we use the "base" model (100M parameters) as the teacher and distill into two smaller "small" (44M) and "xsmall" (22M) variants as students. For Qwen2.5, we use Qwen2.5-1.5B as the teacher and distill into the smaller Qwen2.5-0.5B. Full training details are in Appendix C.

Table 1: **Classification accuracy ($\pm$ std) across datasets with varying total training sizes $k$.** For CoD, training data consists of $k/2$ standard and $k/2$ CFEs. Teacher model `DeBERTa-v3-base` and student model `DeBERTa-v3-small`.

| Dataset | Method | Total Samples ($k$) | | | | | |
| | | 8 | 16 | 32 | 64 | 128 | 512 |
|---------|--------|------|------|------|------|------|------|
| Amazon Polarity | KD | 0.671 ±0.046 | 0.712 ±0.033 | 0.758 ±0.032 | 0.789 ±0.022 | 0.823 ±0.016 | 0.846 ±0.007 |
| | +CoD | **0.758** ±0.027 | **0.795** ±0.033 | **0.819** ±0.035 | **0.812** ±0.004 | **0.837** ±0.014 | **0.860** ±0.015 |
| | LWD | 0.676 ±0.090 | 0.738 ±0.033 | 0.777 ±0.009 | 0.809 ±0.015 | **0.827** ±0.025 | **0.842** ±0.019 |
| | +CoD | **0.724** ±0.052 | **0.779** ±0.056 | **0.811** ±0.015 | **0.828** ±0.015 | 0.816 ±0.020 | 0.841 ±0.013 |
| CoLA | KD | 0.693 ±0.062 | 0.707 ±0.029 | 0.721 ±0.012 | 0.747 ±0.005 | 0.758 ±0.009 | 0.771 ±0.003 |
| | +CoD | **0.739** ±0.026 | **0.755** ±0.017 | **0.769** ±0.011 | **0.769** ±0.016 | **0.772** ±0.006 | **0.791** ±0.004 |
| | LWD | 0.713 ±0.031 | 0.698 ±0.037 | 0.731 ±0.021 | 0.744 ±0.007 | 0.750 ±0.018 | 0.761 ±0.011 |
| | + CoD | **0.730** ±0.035 | **0.744** ±0.031 | **0.762** ±0.011 | **0.752** ±0.009 | **0.756** ±0.010 | **0.784** ±0.003 |
| IMDB | KD | 0.714 ±0.047 | 0.817 ±0.028 | 0.875 ±0.027 | 0.896 ±0.008 | **0.912** ±0.009 | **0.917** ±0.006 |
| | + CoD | **0.835** ±0.078 | **0.888** ±0.005 | **0.890** ±0.011 | **0.899** ±0.007 | 0.907 ±0.006 | 0.913 ±0.005 |
| | LWD | 0.760 ±0.046 | 0.836 ±0.045 | 0.875 ±0.024 | 0.889 ±0.013 | 0.905 ±0.008 | **0.914** ±0.006 |
| | + CoD | **0.861** ±0.017 | **0.886** ±0.011 | **0.893** ±0.006 | **0.898** ±0.005 | 0.905 ±0.010 | 0.913 ±0.010 |
| SST2 | KD | 0.617 ±0.042 | 0.712 ±0.052 | 0.757 ±0.063 | 0.820 ±0.019 | 0.848 ±0.013 | **0.899** ±0.007 |
| | + CoD | **0.719** ±0.063 | **0.781** ±0.034 | **0.821** ±0.013 | **0.827** ±0.008 | **0.853** ±0.015 | 0.892 ±0.018 |
| | LWD | 0.627 ±0.053 | 0.721 ±0.055 | 0.776 ±0.031 | 0.817 ±0.005 | 0.829 ±0.013 | **0.892** ±0.012 |
| | + CoD | **0.694** ±0.079 | **0.785** ±0.028 | **0.832** ±0.011 | **0.830** ±0.007 | **0.835** ±0.012 | 0.880 ±0.020 |
| Yelp | KD | 0.714 ±0.058 | 0.817 ±0.031 | 0.855 ±0.021 | **0.878** ±0.006 | 0.885 ±0.018 | **0.916** ±0.007 |
| | + CoD | **0.740** ±0.094 | **0.832** ±0.045 | **0.860** ±0.018 | 0.874 ±0.006 | **0.888** ±0.013 | 0.913 ±0.011 |
| | LWD | 0.733 ±0.070 | 0.832 ±0.026 | 0.857 ±0.011 | 0.868 ±0.006 | 0.881 ±0.017 | **0.920** ±0.010 |
| | + CoD | **0.738** ±0.093 | **0.865** ±0.010 | **0.870** ±0.017 | **0.871** ±0.019 | **0.885** ±0.007 | 0.913 ±0.013 |
| Sent140 | KD | 0.580 ±0.039 | 0.597 ±0.042 | 0.645 ±0.023 | 0.690 ±0.035 | 0.752 ±0.011 | **0.802** ±0.006 |
| | + CoD | **0.629** ±0.036 | **0.640** ±0.048 | **0.731** ±0.022 | **0.754** ±0.017 | **0.778** ±0.007 | 0.784 ±0.019 |
| | LWD | 0.581 ±0.041 | 0.593 ±0.039 | 0.665 ±0.027 | 0.708 ±0.029 | **0.751** ±0.009 | **0.785** ±0.019 |
| | + CoD | **0.628** ±0.034 | **0.652** ±0.038 | **0.706** ±0.016 | **0.741** ±0.014 | 0.729 ±0.063 | 0.760 ±0.023 |

**Baselines.** We compare our method against three task-aware knowledge distillation baselines: (i) Standard knowledge distillation (KD) where the student learns from the teacher's soft predictions using KL divergence [71]; (ii) Layer-wise distillation (LWD), which extends KD by additionally aligning the student's intermediate hidden representations with those of the teacher using mean squared error [13]; and (iii) TED (Task-aware Layer-wise Distillation) which incorporates task-specific neural filters at each layer to selectively transfer task-relevant information from teacher to student [12]. All methods are evaluated under $k$-shot training settings, and student models are trained on identical few-shot splits to ensure a fair comparison (see details in Appendix C).

**Setup.** As in prior works on task-aware distillation [12], we first train a teacher model on the full training dataset to serve as a strong source of supervision. A student model is then initialized and distilled using only $k$ datapoints, where $k \in \{8, 16, 32, 64, 128, 512\}$. We apply our strategy CoD to three standard distillation baselines: KD, LWD, and TED. For a fair comparison, CoD uses $k/2$ original samples and their $k/2$ corresponding CFE (a total of $k$ shots) while the baseline methods are trained on $k$ original samples. Performance is evaluated using accuracy on the test set for each dataset. All experimental results are averaged over five runs, with the mean and standard deviation reported. Results for the `DeBERTa-v3-base` teacher and `DeBERTa-v3-small` student are shown in Table 1, while results for the smaller `DeBERTa-v3-xsmall` student are in Appendix C. For experiments using the `Qwen2.5-1.5B` teacher and the `Qwen2.5-0.5B` student, see Table 3. We report the accuracy of teacher models trained on the full datasets in Table 4 in Appendix C.

**Results and Analysis.** Across all datasets, we observe that CoD significantly improves performance in the low-data regime, particularly when $k \leq 64$. For example, on `Amazon Polarity` with only 8 labeled examples, KD + CoD achieves 75.8% accuracy compared to 67.1% for standard KD (8.7

Table 2: **Classification accuracy ($\pm$ std) with TED and TED + CoD across datasets and varying total training sizes $k$.** For CoD, training data consists of $k/2$ standard and $k/2$ CFEs. Teacher model is `DeBERTa-v3-base` and student model is `DeBERTa-v3-small`.

| Dataset | Method | Total Samples ($k$) | | | | | |
|---|---|---|---|---|---|---|---|
| | | 8 | 16 | 32 | 64 | 128 | 512 |
| Amazon Polarity | TED | 0.646 ±0.075 | 0.697 ±0.033 | 0.758 ±0.012 | 0.816 ±0.023 | 0.814 ±0.020 | 0.846 ±0.025 |
| | + CoD | **0.731** ±0.054 | **0.754** ±0.056 | **0.802** ±0.007 | **0.818** ±0.013 | **0.805** ±0.008 | **0.848** ±0.010 |
| CoLA | TED | **0.750** ±0.022 | 0.737 ±0.028 | 0.731 ±0.020 | 0.746 ±0.011 | 0.760 ±0.011 | 0.772 ±0.010 |
| | + CoD | 0.748 ±0.028 | **0.757** ±0.023 | **0.767** ±0.021 | **0.768** ±0.016 | **0.780** ±0.007 | **0.791** ±0.006 |
| IMDB | TED | 0.695 ±0.018 | 0.800 ±0.042 | 0.854 ±0.023 | 0.876 ±0.012 | **0.908** ±0.009 | **0.917** ±0.006 |
| | + CoD | **0.827** ±0.056 | **0.879** ±0.003 | **0.884** ±0.007 | **0.887** ±0.010 | 0.895 ±0.010 | 0.916 ±0.005 |
| SST2 | TED | 0.597 ±0.052 | 0.701 ±0.055 | 0.732 ±0.026 | 0.812 ±0.026 | 0.829 ±0.002 | **0.904** ±0.006 |
| | + CoD | **0.658** ±0.087 | **0.779** ±0.012 | **0.813** ±0.017 | **0.833** ±0.014 | **0.836** ±0.030 | 0.879 ±0.011 |
| Yelp | TED | 0.699 ±0.048 | 0.815 ±0.014 | 0.846 ±0.020 | 0.869 ±0.012 | **0.894** ±0.009 | **0.914** ±0.012 |
| | + CoD | **0.742** ±0.095 | **0.837** ±0.016 | **0.868** ±0.018 | **0.878** ±0.019 | 0.886 ±0.013 | 0.913 ±0.008 |

points improvement). Similarly, for `IMDB` at $k = 8$, LWD + CoD improves over standard LWD by more than 10 points (86.1% vs. 76.0%). As the number of labeled examples increases, the benefits of CFE augmentation diminish. At $k = 512$, the performance of standard and CoD becomes nearly identical in many cases. However, even in these larger settings, it is important to note that our method achieves comparable results while using only $k/2$ real samples and $k/2$ CFE, effectively halving the amount of labeled data required to reach similar performance. The effectiveness of CFEs varies by dataset. On `CoLA`, we observe consistent improvements across all $k$ values for both KD and LWD, indicating that CFEs are well-aligned with the task's grammaticality decision boundary. In contrast, datasets like `Sentiment140` show strong early gains. For datasets such as `IMDB` and `SST2`, CFE provides substantial improvements at low $k$, but underperforms slightly at $k = 512$, possibly due to redundancy. Among distillation methods, LWD generally performs on par with or slightly better than KD across most settings, with CoD offering similar relative improvements for both.

We also compare with TED which has been found to work well with larger distillation datasets [12]. We note that TED introduces additional complexity by requiring the training of task-specific filters prior to distillation. Interestingly, we find that TED does not consistently outperform classical methods like KD or LWD in the *few-shot* settings (see Table 2). Nonetheless, TED + CoD yields consistent gains over standard TED, demonstrating that our approach is broadly applicable. Our findings suggest that *simpler distillation approaches like KD or LWD are preferable when data is scarce*: they are easier to implement and, when combined with CoD, deliver much stronger performance gains without the overhead of filter training.

**Ablations.** (1) *On template designing and prompt choices*: We vary prompt templates for generating CFEs and observed that CoD is robust to prompt choices, showing low standard deviation across variants and consistently outperforming the KD baseline few shot settings (see Table 7). One possible direction could be to use automatic prompt generation methods [72], however these are typically more compute-intensive. (2) *Computational and Memory Requirements*: We assess the computational efficiency of CoD under varying few-shot budgets $k$ using the `codecarbon` package [73] to track runtime and energy consumption (see Table 8). (3) *Effect of soft-label supervision*: To study the role of soft-labels in our approach, we remove or corrupting the teacher's soft labels (see Table 9). Removing the soft label term ($\alpha = 0$) leads to a substantial drop in performance across all shot levels. Although CoD still improves over KD in this setting, the gains are significantly reduced. This highlights that the soft label calibration from the teacher is a key contributor to the effectiveness of counterfactual explanation data. Additionally, when replacing soft labels with random values, performance degrades sharply, likely due to inconsistency with the hard labels, introducing conflicting supervision signals in the training objective.

**Discussion.** In this paper, we introduced a novel approach for task-aware knowledge distillation in few-shot settings that leverages CFEs to enhance the data efficiency of knowledge distillation. Our results show that CoD consistently outperforms existing distillation approaches in low-data regimes.

Table 3: **Classification accuracy ($\pm$ std) of** `Qwen2.5` **on** `CoLA` **and** `Yelp` **datasets with varying training sizes $k$.** For CoD training data consists of $k/2$ standard and $k/2$ CFEs. Teacher model is `Qwen2.5-1.5B` and student model is `Qwen2.5-0.5B`. Refer to Appendix C for other datasets.

| Dataset | Method | Total Samples ($k$) | | | | | |
|---|---|---|---|---|---|---|---|
| | | 8 | 16 | 32 | 64 | 128 | 512 |
| CoLA | KD | 0.681 ±0.012 | 0.676 ±0.023 | 0.668 ±0.042 | 0.654 ±0.032 | 0.676 ±0.020 | 0.732 ±0.014 |
| | + CoD | **0.683** ±0.016 | **0.686** ±0.018 | **0.697** ±0.015 | **0.711** ±0.020 | **0.736** ±0.017 | **0.757** ±0.011 |
| | LWD | 0.681 ±0.012 | 0.657 ±0.031 | 0.678 ±0.018 | 0.650 ±0.039 | 0.636 ±0.029 | 0.712 ±0.014 |
| | + CoD | **0.682** ±0.018 | **0.687** ±0.013 | **0.704** ±0.010 | **0.714** ±0.020 | **0.719** ±0.022 | **0.755** ±0.013 |
| Yelp | KD | 0.684 ±0.021 | 0.759 ±0.040 | 0.827 ±0.030 | 0.861 ±0.017 | **0.887** ±0.012 | **0.920** ±0.010 |
| | + CoD | **0.745** ±0.029 | **0.779** ±0.048 | **0.828** ±0.072 | **0.886** ±0.007 | 0.883 ±0.010 | 0.916 ±0.008 |
| | LWD | 0.685 ±0.019 | 0.777 ±0.036 | 0.837 ±0.027 | 0.876 ±0.020 | **0.898** ±0.008 | **0.920** ±0.005 |
| | + CoD | **0.746** ±0.028 | **0.778** ±0.035 | **0.847** ±0.020 | **0.876** ±0.014 | 0.883 ±0.010 | 0.909 ±0.009 |

Importantly, we demonstrate that CoD can achieve improved performance over baselines while effectively using only half the number of original data, with the remainder consisting of generated CFEs. This finding has significant implications for reducing the cost of data collection in real-world scenarios where sourcing high-quality data is expensive or time-consuming [74]. Our approach offers an explanation-driven perspective on distillation. By including CFE's, we implicitly highlight the key features most important to flipping a teacher's decision. This may help the student model reduce its reliance on spurious correlations, especially in few-shot settings. In effect, CFE's guide the student to attend to "why" a label changes, not just "what" the label is. This bridges explainability and compression, turning explanations into actionable data for knowledge distillation.

**On the extension to generative LLMs.** As research increasingly focuses on data-efficient LLM training [75, 76], future work could extend our approach to generative sequence-to-sequence models, enabling efficient distillation beyond classification. In this setting, a CFE can be defined as a minimal change to the input (prompt) that flips a chosen property of the generated text. Formally, given a generative model $f$, a prompt $\mathbf{x}$, and a binary attribute function $g(\text{output})$ (e.g., sentiment, toxicity, factuality, topic relevance), a counterfactual explanation prompt $\mathbf{x}^\star$ would be a small semantic perturbation of $x$ such that the generated output $f(\mathbf{x}^\star)$ flips the value of $g(f(\mathbf{x}))$. More broadly, CFEs can be reframed through *model sensitivity*, identifying small input perturbations that cause large changes in the output distribution or likelihood. This corresponds to large changes in the sequence-level probability: $p_\theta(y \mid \mathbf{x}) = \prod_{t=1}^{T} p_\theta(y_t \mid \mathbf{x}, y_{<t})$, where $y = (y_1, y_2, \ldots y_T)$ is the generated sequence for an input prompt $\mathbf{x}$. This could reveal regions of the input space where the model is uncertain, making them especially informative for distillation, supervised fine-tuning, or post-training. Our approach offers a path toward data-efficient LLM distillation from minimal data while reducing cost and maintaining performance.

**Limitations.** Generating CFEs introduces additional computational overhead compared to standard distillation approaches. Moreover, our current CFE generation strategy, which relies on prompting LLMs, does not guarantee that we would get the closest counterfactual (as defined in Definition 1), potentially limiting the precision of our distilled knowledge. Future work could explore alternate methods for generating closer and semantically valid CFEs. Additionally, as with knowledge distillation in general, CoD is inherently dependent on the quality of the teacher model. Any inaccuracies or biases present in the teacher's decision boundary may be inherited by the student. Addressing robustness to flawed teachers remains an important direction for future research.

**Societal Impact.** CoD offers several potential societal impacts, particularly in reducing the cost and effort associated with data collection [74]. By enabling the distillation of high-performance models with fewer data samples, this approach can significantly lower data collection costs, making machine learning more accessible in low-resource environments. This is especially valuable in industries where data is often scarce and expensive to obtain [77, 74]. Moreover, by requiring fewer samples and targeting smaller student models, CoD contributes to more efficient model training and scalable deployment. Our method leverages explanations as a tool for more effective model compression. In doing so, it bridges the gap between explainability and model compression.

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

# Appendix

## A  Background on Fisher Information and Proof of Theorem 1

This section provides background on Fisher information and a formal proof for Theorem 1, which quantifies the reduction in estimation error from using CFE-infused training data.

### A.1  Background on Fisher Information Matrix

**Definition 4** (Positive Semi-definite Matrices). *A matrix $A \in \mathbb{R}^{d \times d}$ is said to be positive semi-definite if it is symmetric and for all non-zero vectors $\mathbf{x} \in \mathbb{R}^d$, the following condition holds:*

$$\mathbf{x}^T A \mathbf{x} \geq 0 \quad \text{for all} \quad \mathbf{x} \in \mathbb{R}^d.$$

*The eigenvalues of a positive semi-definite are non-negative, i.e., $\lambda_i(A) \geq 0$ for all eigenvalues $\lambda_i$ of $A$.*

**Definition 5** (Löwner Order). *Let $A, B \in \mathbb{R}^{d \times d}$ be symmetric matrices. We say that $A$ is greater than or equal to $B$ in the Löwner order, denoted $A \succeq B$, if and only if the matrix $A - B$ is positive semi-definite. That is,*

$$A \succeq B \quad \text{if and only if} \quad x^T (A - B) x \geq 0 \quad \text{for all} \quad x \in \mathbb{R}^d.$$

*If $A \succ B$, then $A - B$ is positive definite, meaning $A$ is strictly greater than $B$ in the Löwner order.*

**Lemma 2** (Trace Inequality for Positive Semi-definite Matrices). *For positive semi-definite matrices $A, B \in \mathbb{R}^{d \times d}$ where $A \succ B$, then:*

$$Tr(A^{-1}) < Tr(B^{-1})$$

*Proof.* Since $A \succ B$, we have $B^{-1} \succ A^{-1}$ by the Löwner order inversion property. The trace operator preserves this inequality because for any $X \succ Y \succ 0$:

$$\text{Tr}(X) = \sum_{i=1}^{d} \lambda_i(X) > \sum_{i=1}^{d} \lambda_i(Y) = \text{Tr}(Y)$$

where $\lambda_i(\cdot)$ denotes eigenvalues in descending order. $\square$

**Definition 2** (Fisher Information Matrix [59]). *Let $\mathcal{L}(\theta)$ be the log-likelihood of a parametric distribution $p(y, x; \theta)$, where $\theta$ is the parameter vector to be estimated. The Fisher Information Matrix (FIM) at parameter $\theta$ is defined as:*

$$\mathcal{I}(\theta) = \mathbb{E}_{\mathbf{x}, y} \left[ \nabla_\theta \log p(y, \mathbf{x}; \theta) \, \nabla_\theta \log p(y, \mathbf{x}; \theta)^\top \right].$$

Fisher information captures the amount of information that an observable random variable $x$ carries about an unknown parameter $\theta$ of a distribution that models $x$. We use the notation $\mathcal{I}(\theta; y, \mathbf{x})$ to denote the Fisher information about $\theta$ carried by single observation $y, \mathbf{x}$.

### A.2  Proof of Theorem 1

**Theorem 1** (CFEs Improve Model Parameter Estimation). *Let $\mathbf{w}_s$ and $\mathbf{w}_s^{(\text{cf})}$ be the student parameters obtained via MLE on $\mathcal{D}$ (standard) and $\mathcal{D}_{\text{cf}}$ (CFE-infused). Assuming the teacher's parameters $\mathbf{w}_t$ capture the true data-generating distribution, that CFEs lie near the decision boundary, and that the second moments $\mathbb{E}_{\mathbf{x}}[\mathbf{x}\mathbf{x}^\top] \approx \mathbb{E}_{\mathbf{x}_c}[\mathbf{x}_c \mathbf{x}_c^\top]$. Then estimation error satisfies:*

$$\mathbb{E}\left[ \|\mathbf{w}_s^{(\text{cf})} - \mathbf{w}_t\|^2 \right] < \mathbb{E}\left[ \|\mathbf{w}_s - \mathbf{w}_t\|^2 \right].$$

*Proof.* For a single observation $(\mathbf{x}, y)$, the log-likelihood is:

$$\log p(y|\mathbf{x}; \mathbf{w}) = y \log \sigma(\mathbf{w}^\top \mathbf{x}) + (1 - y) \log(1 - \sigma(\mathbf{w}^\top \mathbf{x})) \tag{3}$$

Taking the gradient with respect to $\mathbf{w}$:

$$\nabla_{\mathbf{w}} \log p(y|\mathbf{x}; \mathbf{w}) = (y - \sigma(\mathbf{w}^\top \mathbf{x}))\mathbf{x} \tag{4}$$

To prove Theorem 1, we first (1) Characterize the Fisher information for individual observations, (2) Establish asymptotic normality of MLE, (3) Compare information matrices of standard vs. CFE-infused datasets, and (4) Apply trace inequality to connect information to estimation error.

*(1) Fisher Information for Logistic Regression*: For a logistic regression model with parameters $\mathbf{w}$, Lets denote Fisher Information Matrix (FIM) for observations $y, \mathbf{x}$ as:

$$\mathcal{I}(\mathbf{w}; y, \mathbf{x}) = \mathbb{E}_{y,\mathbf{x}} \left[ \nabla_{\mathbf{w}} \log p(y, \mathbf{x}; \mathbf{w}) \nabla_{\mathbf{w}} \log p(y, \mathbf{x}; \mathbf{w})^{\top} \right] \tag{5}$$

$$\nabla_{\mathbf{w}} \log p(y, \mathbf{x}; \mathbf{w}) = \nabla_{\mathbf{w}} \log p(y|\mathbf{x}; \mathbf{w}) + \underbrace{\nabla_{\mathbf{w}} \log p(\mathbf{x})}_{=0}. \tag{6}$$

The gradient of $\log p(\mathbf{x})$ is zero because $p(\mathbf{x})$ is independent of the model parameters $\mathbf{w}$.

Using the law of total expectation:

$$\mathcal{I}(\mathbf{w}; y, \mathbf{x}) = \mathbb{E}_{\mathbf{x}}[\mathbb{E}_{y|\mathbf{x}} \left[ \nabla_{\mathbf{w}} \log p(y|\mathbf{x}; \mathbf{w}) \nabla_{\mathbf{w}} \log p(y|\mathbf{x}; \mathbf{w})^{\top} \right] \tag{7}$$

Substituting Equation 4:

$$\mathcal{I}(\mathbf{w}; y, \mathbf{x}) = \mathbb{E}_{\mathbf{x}} \left[ \mathbb{E}_{y|\mathbf{x}}[(y - \sigma(\mathbf{w}^{\top}\mathbf{x}))^2 \mathbf{x}\mathbf{x}^{\top}]] \right] \tag{8}$$

$$= \mathbb{E}_{\mathbf{x}} \left[ \mathbf{x}\mathbf{x}^{\top} \mathbb{E}_{y|\mathbf{x}}[(y - \sigma(\mathbf{w}^{\top}\mathbf{x}))^2]] \right] \tag{9}$$

The term $\mathbb{E}_{y|\mathbf{x}} \left[ (y - \sigma(\mathbf{w}^{\top}\mathbf{x}))^2 \right]$ is the variance of $y|\mathbf{x}$. Where $y|\mathbf{x} \sim \text{Bernoulli}(\sigma(\mathbf{w}^{\top}\mathbf{x}))$, we compute:

$$\mathbb{E}_{y|\mathbf{x}}[(y - \sigma(\mathbf{w}^{\top}\mathbf{x}))^2] = \text{Var}(y|\mathbf{x}) = \sigma(\mathbf{w}^{\top}\mathbf{x})(1 - \sigma(\mathbf{w}^{\top}\mathbf{x})) \tag{10}$$

Thus:

$$\mathcal{I}(\mathbf{w}; y, \mathbf{x}) = \mathbb{E}_{\mathbf{x}}[\sigma(\mathbf{w}^{\top}\mathbf{x})(1 - \sigma(\mathbf{w}^{\top}\mathbf{x}))\mathbf{x}\mathbf{x}^{\top}] \tag{11}$$

The variance term is maximized when $\mathbf{w}^{\top}\mathbf{x} = 0$ (i.e., at the decision boundary), where it equals 0.25.

*(2) Asymptotic Distribution of MLE*: Under regularity conditions [77], the MLE estimator satisfies:

$$\sqrt{k}(\mathbf{w}_s - \mathbf{w}_t) \xrightarrow{d} \mathcal{N}(0, \mathcal{I}^{-1}(\mathbf{w}_t; \mathcal{D})) \tag{12}$$

where $\mathcal{I}(\mathbf{w}_t; \mathcal{D}) = \sum_{i=1}^{k} \mathcal{I}(\mathbf{w}_t; y_i, \mathbf{x}_i)$ is the total Fisher information of $k$ independent observations of $y_i, \mathbf{x}_i$ (Additivity property of fisher information [78]).

The mean squared error (MSE) [79] decomposes as:

$$\mathbb{E}\|\mathbf{w}_s - \mathbf{w}_t\|^2 = \underbrace{\text{Tr}(\text{Cov}(\mathbf{w}_s))}_{\text{Variance}} + \underbrace{\|\text{Bias}(\mathbf{w}_s)\|^2}_{\text{Bias}} \tag{13}$$

For MLE, $\text{Bias}(\mathbf{w}_s) \to 0$ as $k \to \infty$, so: $\mathbb{E}\|\mathbf{w}_s - \mathbf{w}_t\|^2 \approx \text{Tr}(\mathcal{I}^{-1}(\mathbf{w}_t; \mathcal{D}))$

The next step of the proof we compare the fisher information between a standard dataset and CFE-infused dataset.

Let $\mathcal{D} = \{\mathbf{x}_i\}_{i=1}^{k}$ be a dataset of $k$ standard samples, and let $\mathcal{D}_{\text{cf}} = \{\mathbf{x}_i\}_{i=1}^{k/2} \cup \{\mathbf{x}_{c_j}\}_{j=1}^{k/2}$ be an CFE-infused dataset containing $k/2$ standard samples and $k/2$ CFEs.

Standard Samples: Far from decision boundary $\Rightarrow \mathbf{w}_t^{\top}\mathbf{x}_i \gg 0$ or $\ll 0$. Thus:

$$\sigma(\mathbf{w}_t^{\top}\mathbf{x}_i)(1 - \sigma(\mathbf{w}_t^{\top}\mathbf{x}_i)) = \epsilon_i < 0.25 \tag{14}$$

Their FIM contribution is: $\mathcal{I}(\mathbf{w}_t; \mathbf{x}_i) = \mathbb{E}_{\mathbf{x}}[\epsilon_i \mathbf{x}_i \mathbf{x}_i^{\top}]$.

CFE Samples: Near boundary $\Rightarrow \mathbf{w}_t^{\top}\mathbf{x}_c = 0 \Rightarrow \sigma(0) = 0.5$. Thus:

$$\sigma(\mathbf{w}_t^{\top}\mathbf{x}_c)(1 - \sigma(\mathbf{w}_t^{\top}\mathbf{x}_c)) = 0.25 \tag{15}$$

Their FIM contribution is maximal: $\mathcal{I}(\mathbf{w}_t; \mathbf{x}_c) = \mathbb{E}_{\mathbf{x}}[0.25\mathbf{x}_c \mathbf{x}_c^{\top}]$.

Since $\mathbb{E}_{\mathbf{x}}[\mathbf{x}\mathbf{x}^{\top}] \approx \mathbb{E}_{\mathbf{x}_c}[\mathbf{x}_c \mathbf{x}_c^{\top}]$ and $0.25 \gg \epsilon_i$, we have $\mathcal{I}(\mathbf{w}_t; \mathcal{D}_{cf}) \succ \mathcal{I}(\mathbf{w}_t; \mathcal{D})$ in the Löwner order (see Definition 5).

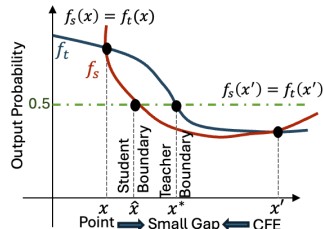

Figure 5: Intuition for Theorem 2

**Remark 2** (Feature Spanning). *Note that for logistic regression the feature vector is augmented with the parameter bias term, i.e., $\mathbf{x} = [1, \tilde{\mathbf{x}}^\top]^\top$, hence, the outer product $\mathbf{x}\mathbf{x}^\top$ has a non-zero norm. The first element of $\mathbf{x}$ is always 1, ensuring $\|\mathbf{x}\|^2 \geq 1$. Thus, $\mathbf{x}\mathbf{x}^\top$ cannot be the zero matrix, even if $\hat{\mathbf{x}} = \mathbf{0}$. This guarantees that each CFE example $\mathbf{x}_c$ contributes a non-degenerate rank-1 term to the FIM.*

The final step of the proof leverages the trace inequality for covariance matrices (see Lemma 2). If $\mathcal{I}(\mathbf{w}_t; \mathcal{D}_{cf}) \succ \mathcal{I}(\mathbf{w}_t; \mathcal{D})$ then $\mathrm{Tr}(\mathcal{I}^{-1}(\mathbf{w}_t; \mathcal{D}_{cf})) < \mathrm{Tr}(\mathcal{I}^{-1}(\mathbf{w}_t; \mathcal{D}))$. Thus, CFE infusion reduces parameter estimation error:

$$\mathbb{E}\left[\|\mathbf{w}_s^{(\mathrm{cf})} - \mathbf{w}_t\|^2\right] < \mathbb{E}\left[\|\mathbf{w}_s - \mathbf{w}_t\|^2\right] \tag{16}$$

**Remark 3** (Datapoint Diversity). *For the total FIM $\mathcal{I}(\mathbf{w}_t; \mathcal{D}_{cf})$ to be invertible, the set of feature vectors $\{\mathbf{x}_i\}$ must span $\mathbb{R}^d$ which will hold if we have enough samples.*

$\square$

# B  Background on Hausdorff Distance and Proofs of Lemma 1 and Theorem 2

This section provides definitions and geometric preliminaries, along with proofs for Lemma 1 and Theorem 2.

## B.1  Background on Hausdorff Distance

**Definition 6** (Line Segment). *Let $\mathbf{x}_i, \mathbf{x}_i' \in \mathbb{R}^{n \times d}$ be two points in the $n \times d$ space. The line segment $[\mathbf{x}_i, \mathbf{x}_i']$ connecting $\mathbf{x}_i$ and $\mathbf{x}_i'$ is defined as the set of points $\gamma(\lambda)$ for $\lambda \in [0, 1]$, where*

$$\gamma(\lambda) = (1 - \lambda)\mathbf{x}_i + \lambda\mathbf{x}_i', \quad \lambda \in [0, 1].$$

*This defines all the points on a space between $\mathbf{x}_i$ and $\mathbf{x}_i'$ in $\mathbb{R}^{n \times d}$.*

**Lemma 3** (Intermediate Value Theorem). *Let $f : [a, b] \to \mathbb{R}$ be a continuous function, and let $f(a) \neq f(b)$. If $y$ is any value between $f(a)$ and $f(b)$, then there exists $c \in (a, b)$ such that $f(c) = y$.*

**Definition 3** (Hausdorff Distance). *Let $\mathcal{M}_t, \mathcal{M}_s \subseteq \mathbb{R}^{n \times d}$ be two non-empty subsets of a metric space. The* Hausdorff distance *is defined as:*

$$\mathrm{H}(\mathcal{M}_s, \mathcal{M}_t) = \max\left\{\sup_{\mathbf{x} \in \mathcal{M}_t} \inf_{\mathbf{u} \in \mathcal{M}_s} \|\mathbf{x} - \mathbf{u}\|_F, \sup_{\mathbf{u} \in \mathcal{M}_s} \inf_{\mathbf{x} \in \mathcal{M}_t} \|\mathbf{u} - \mathbf{x}\|_F\right\}.$$

## B.2  Proofs of Lemma 1 and Theorem 2

**Lemma 1** (Existence of Boundary Crossing for Counterfactual Pairs). *Let $f_t : \mathbb{R}^{n \times d} \to [0, 1]$ be a continuous function. For a datapoint and its counterfactual pair $(\mathbf{x}_i, \mathbf{x}_i')$, there exists a point $\mathbf{x}_i^\star = \alpha\mathbf{x}_i + (1 - \alpha)\mathbf{x}_i'$ for an $\alpha \in (0, 1)$ (on the line joining $\mathbf{x}_i$ and $\mathbf{x}_i'$) such that: $f_t(\mathbf{x}_i^\star) = 0.5$.*

*Proof.* Define the line segment from $x_i$ to $x_i'$ using a parameterization: $\gamma(\lambda) = (1 - \lambda)x_i + \lambda x_i'$, for $\lambda \in [0, 1]$.

This defines a continuous path from $x_i$ to $x_i'$ in $\mathbb{R}^d$. Now define the real-valued function $g : [0, 1] \to \mathbb{R}$ by: $g(\lambda) = f_t(\gamma(\lambda)) = f_t((1 - \lambda)x_i + \lambda x_i')$.

Since $f_t$ is continuous on $\mathbb{R}^d$, and $\gamma(\lambda)$ is continuous in $\lambda$, the composition $g(\lambda)$ is continuous on the closed interval $[0, 1]$.

Now, evaluate the endpoints of this function: $g(0) = f_t(x_i) < 0.5, g(1) = f_t(x_i') > 0.5$.

Thus, we have $g(0) < 0.5 < g(1)$, and by the Intermediate Value Theorem (see Lemma 3), since $g$ is continuous on $[0, 1]$, there exists $\lambda^\star \in (0, 1)$ such that: $g(\lambda^\star) = 0.5$.

Define $x_i^\star = \gamma(\lambda^\star) = (1 - \lambda^\star)x_i + \lambda^\star x_i' \in [x_i, x_i']$. Then: $f_t(x_i^\star) = g(\lambda^\star) = 0.5$.

Hence, the point $x_i^\star \in [x_i, x_i']$ lies on the segment and satisfies $f_t(x_i^\star) = 0.5$, as required. $\qquad\square$

**Theorem 2** (Teacher–Student Boundary Proximity). *Let $f_t, \ f_s : \mathbb{R}^{n \times d} \to; [0, 1]$ be the teacher and student model, with decision boundaries $\mathcal{M}_t = \{\mathbf{x} \mid f_t(\mathbf{x}) = 0.5\}$ and $\mathcal{M}_s = \{\mathbf{x} \mid f_s(\mathbf{x}) = 0.5\}$, respectively. Assume we observe a CFE-infused dataset $\mathcal{D}_{cf} = \left\{(\mathbf{x}_i, \mathbf{x}_i')\right\}_{i=1}^{k}$ satisfying: (A1) Minimal perturbation: $\|\mathbf{x}_i - \mathbf{x}_i'\|_F \leq \alpha$ with $\alpha > 0$; (A2) Exact distillation: $f_s(\mathbf{x}_i) = f_t(\mathbf{x}_i)$ and $f_s(\mathbf{x}_i') = f_t(\mathbf{x}_i')$; and (A3) $\varepsilon$-spread along the teacher and student boundary, i.e., for each pair, there exist a teacher's (or student's) crossing point $\mathbf{x}_i^\star = \alpha\mathbf{x}_i + (1 - \alpha)\mathbf{x}_i'$ for $\alpha \in (0, 1)$ such that $f_t(x_i^\star) = 0.5$ (or, $f_s(x_i^\star) = 0.5$) and for every $a \in \mathcal{M}_t$ (or $\mathcal{M}_s$), there exists an $i$ with $\|a - \mathbf{x}_i^\star\|_2 \leq \varepsilon$. Then the Hausdorff distance between the decision boundaries obeys:* $\mathrm{H}(\mathcal{M}_s, \mathcal{M}_t) \leq \alpha + \varepsilon$.

*Proof.* To prove Theorem 2, we bound the Hausdorff distance between the student's and teacher's decision boundaries using the given assumptions. We bound each term separately of the Hausdorff distance (see Defintion 3).

We first bound $\sup_{\mathbf{x} \in \mathcal{M}_t} \inf_{\mathbf{u} \in \mathcal{M}_s} \|\mathbf{x} - \mathbf{u}\|_F$:

For any $a \in \mathcal{M}_t$, by assumption (A3), there exists a CFE pair $(\mathbf{x}_i, \mathbf{x}_i')$ with teacher crossing point $\mathbf{x}_i^\star \in \mathcal{M}_t$ such that:
$$\|a - \mathbf{x}_i^\star\|_F \leq \varepsilon. \tag{17}$$

The segment $[\mathbf{x}_i, \mathbf{x}_i']$ has length $\|\mathbf{x}_i - \mathbf{x}_i'\|_F \leq \alpha$ (A1). By Lemma 1 and (A2) Exact distillation, the student's boundary $\mathcal{M}_s$ intersects $[\mathbf{x}_i, \mathbf{x}_i']$ at some $\mathbf{u}_i^\star \in \mathcal{M}_s$. Since $\mathbf{x}_i^\star$ and $\mathbf{u}_i^\star$ lie on $[\mathbf{x}_i, \mathbf{x}_i']$, their distance satisfies:
$$\|\mathbf{x}_i^\star - \mathbf{u}_i^\star\|_F \leq \|\mathbf{x}_i - \mathbf{x}_i'\|_F \leq \alpha. \tag{18}$$

Combining Equation 18 and 17:
$$\|a - \mathbf{u}_i^\star\|_F \leq \|a - \mathbf{x}_i^\star\|_F + \|\mathbf{x}_i^\star - \mathbf{u}_i^\star\|_F \leq \varepsilon + \alpha. \tag{19}$$

Thus, $\inf_{\mathbf{u} \in \mathcal{M}_s} \|a - \mathbf{u}\|_F \leq \varepsilon + \alpha$. Taking the supremum over $a \in \mathcal{M}_t$:
$$\sup_{\mathbf{x} \in \mathcal{M}_t} \inf_{\mathbf{u} \in \mathcal{M}_s} \|\mathbf{x} - \mathbf{u}\|_F \leq \varepsilon + \alpha. \tag{20}$$

Next we bound $\sup_{\mathbf{u} \in \mathcal{M}_s} \inf_{\mathbf{x} \in \mathcal{M}_t} \|\mathbf{u} - \mathbf{x}\|_F$:

From (A1), the distance between the student's cutpoint $\mathbf{u}_i^*$ and the teacher's cutpoint $\mathbf{x}_i^\star$ satisfies:
$$\|\mathbf{u}_i^* - \mathbf{x}_i^\star\|_F \leq \|\mathbf{x}_i - \mathbf{x}_i'\|_F \leq \alpha \tag{21}$$

For any other $\mathbf{u} \in \mathcal{M}_s$:
$$\|\mathbf{u} - \mathbf{x}_i^\star\|_F \leq \|\mathbf{u} - \mathbf{u}_i^*\|_F + \|\mathbf{u}_i^* - \mathbf{x}_i^\star\|_F \leq \varepsilon + \alpha, \tag{22}$$

Assuming the CFE pairs $(\mathbf{x}_i, \mathbf{x}_i')$ intersection points are $\varepsilon-$spread (well spread) along the student decision boundary.

Since $\mathbf{x}_i^\star \in \mathcal{M}_t$, we have:
$$\inf_{\mathbf{x} \in \mathcal{M}_t} \|\mathbf{u} - \mathbf{x}\|_F \leq \|\mathbf{u} - \mathbf{x}_i^\star\|_F \leq \varepsilon + \alpha. \tag{23}$$

Taking the supremum over $\mathbf{u} \in \mathcal{M}_s$:

$$\sup_{\mathbf{u} \in \mathcal{M}_s} \inf_{\mathbf{x} \in \mathcal{M}_t} \|\mathbf{u} - \mathbf{x}\|_F \leq \varepsilon + \alpha. \tag{24}$$

Combining both bounds, the Hausdorff distance is the maximum of the two suprema:

$$\mathrm{H}(\mathcal{M}_s, \mathcal{M}_t) \leq \max\{\varepsilon + \alpha, \ \varepsilon + \alpha\} = \varepsilon + \alpha \tag{25}$$

$\square$

## C    Additional Experiments and Details

This appendix provides additional experimental details and results to supplement the main paper. In Appendix C.1, we describe the datasets used in our few-shot experiments and preprocessing choices. In Appendix C.2, we include the prompt templates used to generate counterfactual explanations. Baseline methods are summarized in Appendix C.3, and complete hyperparameter settings are detailed in Appendix C.4. Finally, Appendix C.5 presents extended results using the smaller DeBERTa-v3-xsmall student and the Qwen2.5 model family.

### C.1    Datasets Details

We evaluate CoD across six text classification benchmarks that span a range of domains. For each $k$-shot setup, we sample a balanced subset from the processed training data, selecting $k/2$ examples per class. All experiments are repeated across 5 random seeds, each with a different sampled subset.

- Yelp [70]: We use the Yelp Review Full dataset, filtering for reviews with at most 250 tokens and discarding neutral labels. Labels are binarized: 1–2 as negative and 4–5 as positive. The processed dataset contains 106,624 training examples, 1,000 for validation, and 7,074 for testing, with a slightly imbalanced class distribution (64% negative).

- IMDB [67]: We retain only reviews with shorter lengths. The original test and unsupervised splits are repurposed as validation and test sets, respectively. The resulting data includes 782 training, 858 validation, and 1,578 test samples, with the test set unlabeled.

- SST2 2 [65]: We use the full GLUE-provided training, validation, and test splits without modification. The train/val sets contain 67,349 and 872 examples, respectively. The test set has 1,821 unlabeled examples.

- CoLA [68]: We adopt the standard GLUE splits of the CoLA dataset, yielding 8,551 training, 1,043 validation, and 1,063 unlabeled test samples. The task is binary classification of linguistic acceptability.

- Sentiment140 [66]: We filter the dataset to exclude neutral tweets. The final dataset includes 1,598,400 training, 1,600 validation, and 359 test examples, with balanced label distributions.

- Amazon Polarity [69]: We select examples with shortest length. The processed data includes 1,111 training and 113 validation samples, with roughly balanced sentiment labels.

### C.2    Counterfactual Explanation Generation Prompt Templates

Here we provide prompt templates used for counterfactual explanation generation across datasets. Each prompt instructs the model to minimally modify a given input to flip the class label (e.g., sentiment or grammaticality) while preserving meaning and structure. We used gpt-4o-2024-11-20 [64] for our CFE generation.

```
SST-2 / IMDB / Sentiment140 / Amazon
```

You are an AI assistant tasked with generating counterfactual explanations for sentiment analysis.
Given a sentence and its true sentiment label, your goal is to make the minimal necessary change to flip
the sentiment while preserving the structure and meaning as much as possible.
For example, if the input is:
Sentence: "I love this movie."
True sentiment: Positive
A suitable counterfactual explanation would be: "I dislike this movie."
Now, generate a counterfactual explanation for the following sentence:
Sentence: {sentence}
True sentiment: {sentiment}
Return only the counterfactual sentence, without any additional information.

```
Yelp
```

You are an AI assistant tasked with generating counterfactual explanations for sentiment analysis of
Yelp reviews.
Given a sentence (a Yelp review) and its true sentiment label (positive or negative), your goal is to make
the minimal necessary change to flip the sentiment while preserving the structure and meaning as much
as possible.
For example, if the input is:
Sentence: "This restaurant is fantastic, the food was amazing!"
True sentiment: Positive
A suitable counterfactual explanation would be: "This restaurant is terrible, the food was awful!"
Now, generate a counterfactual explanation for the following sentence:
Sentence: {sentence}
True sentiment: {sentiment}
Return only the counterfactual sentence, without any additional information.

```
CoLA
```

You are an AI assistant tasked with generating counterfactual explanations for grammaticality judgment.
Given a sentence and its true grammaticality label (Acceptable or Unacceptable), your goal is to make
the minimal necessary change to flip the grammaticality while preserving the structure and meaning as
much as possible.
For example, if the input is:
Sentence: "She is going to the store."
True grammaticality: Acceptable
A suitable counterfactual explanation would be: "She is go to the store."
Now, generate a counterfactual explanation for the following sentence:
Sentence: {sentence}
True grammaticality: {sentiment}
Return only the counterfactual sentence, without any additional information.

### C.3 Baselines Details

We compare CoD against three task-aware knowledge distillation methods widely used for distillation.
CoD uses $k/2$ original samples and their $k/2$ corresponding CFE (a total of $k$ shots) while the baseline
methods are trained on $k$ original samples.

- **Knowledge Distillation (KD)** [71]: A classical distillation approach where the student model
  learns to mimic the teacher's soft target probabilities using Kullback-Leibler (KL) divergence. This
  method transfers predictive behavior but does not supervise intermediate representations.

- **Layer-wise Distillation (LWD)** [13]: An extension of KD that additionally aligns the student's
  intermediate hidden representations with those of the teacher. This is typically done via a mean
  squared error loss over corresponding layers, encouraging the student to internalize not only the
  final outputs but also the hierarchical feature representations of the teacher.

Table 4: **Teacher accuracy (%) across datasets.** Reporting `Qwen2.5-1.5B` and `DeBERTa-v3-base` when fine-tuned on full training dataset for each benchmark. These teachers are used as sources of supervision for student models during knowledge distillation.

| Model | Amazon Polarity | CoLA | IMDB | SST2 | Yelp | Sentiment140 |
|---|---|---|---|---|---|---|
| Qwen2.5-1.5B | 88.5 | 83.0 | 94.3 | 93.7 | 95.4 | 86.1 |
| DeBERTa-v3-base | 86.7 | 87.5 | 93.8 | 95.8 | 95.6 | 86.8 |

- **Task-aware Layer-wise Distillation (TED)** [12]: TED augments LWD with learned neural filters at each layer of both teacher and student models. These filters are trained to select task-relevant information from intermediate representations before computing the distillation loss. This selective transfer enables more effective compression by focusing on information critical to task performance.

## C.4 Models and Hyperparameters

- **DeBERTa-V3 [25].** We fine-tune the teacher model using DeBERTaV3-base, initialized with a classification head for each target task. For the teacher, we use a dropout rate of $0.1$, linear learning rate decay, and train for $8$ epochs with a fixed learning rate of $2 \times 10^{-5}$ and batch sizes of $\{32, 64\}$. Optimization is performed using Adam with $\epsilon = 1 \times 10^{-6}$, $\beta_1 = 0.9$, and $\beta_2 = 0.98$, without weight decay. Mixed-precision training with FP16 is used throughout.

  For distillation, the student is initialized from a pre-trained `DeBERTa-v3-small` or `DeBERTa-v3-xsmall` model. We search learning rates in the range $[1 \times 10^{-5}, 5 \times 10^{-5}]$, and use a fixed batch size of $8$ in our few-shot experiments. All student models are trained for $10$ epochs using Adam with the same optimizer settings as the teacher. For KD and LWD baselines, we set the distillation loss weight to $20$. For the TED baseline, we use the same hyperparameters for both the filter training and distillation phases, consistent with [12].

- **Qwen2.5 [2].** We use `Qwen/Qwen2.5-1.5B` as the teacher and `Qwen/Qwen2.5-0.5B` as the student, both loaded from Hugging Face with sequence classification heads. We fine-tune using a batch size of 16 and train for 10 epochs. For KD and LWD baselines, we set the distillation loss weights to 20 and 5, respectively. All other settings closely follow the DeBERTaV3 setup, including the optimizer, learning rate schedule, and use of mixed-precision training.

  All experiments are conducted on a server equipped with four NVIDIA RTX A6000 GPUs.

## C.5 Additional Results and Discussion.

We provide results using the smaller `DeBERTa-v3-xsmall` (22M parameters) student as well as the full evaluation table for the `Qwen2.5` family. Results for the smaller `DeBERTa-v3-xsmall` student are shown in Table 5. While experiments using the `Qwen2.5-1.5B` teacher and the `Qwen2.5-0.5B` student are shown in Table 6. We also include the full fine-tuned teacher model accuracies across all datasets in Table 4, which are used as supervision targets during knowledge distillation. All experimental results are averaged over five runs, with the mean and standard deviation reported.

Overall, our findings corroborate the central insight that infusing CFEs into knowledge distillation significantly boosts model performance in few-shot settings. For the smaller `DeBERTa-v3-xsmall` student, we observe that the benefits of CFE infusion remain substantial across tasks, especially when $k \leq 64$. For example, on IMDB at $k = 8$, KD + COD improves from 74.3% to 89.3%, and LWD + COD improves from 77.3% to 87.7%, showing that even with a much smaller student, CFEs offer a powerful training signal. Similar patterns are seen on SST2 and `Amazon Polarity`. While the performance gap narrows at higher $k$ values, our method still matches or slightly outperforms standard distillation, despite using only half as many real samples. These results highlight the scalability of COD across student model sizes.

We also evaluate COD on `Qwen2.5` models, using `Qwen2.5-1.5B` as the teacher and `Qwen2.5-0.5B` as the student. Results on CoLA, Yelp, `Amazon Polarity`, and IMDB show that our method consistently outperforms standard KD and LWD, particularly in few-shot regimes. On IMDB with $k=8$, KD + COD reaches 80.0% vs. 67.8% for standard KD - a remarkable 12.2 percentage point gain. Similarly, LWD + COD improves CoLA accuracy by 8.3 points at $k=128$ (71.9% vs. 63.6%). With ($k=8$), COD boosts Yelp performance by 6.1 points for both KD (74.5% vs. 68.4%) and LWD (74.6%

Table 5: **Classification accuracy ($\pm$ std) across datasets with varying total training sizes** $k$. For CoD, training data consists of $k/2$ standard and $k/2$ CFEs. Teacher model `DeBERTa-v3-base` and student model `DeBERTa-v3-xsmall`

| Dataset | Method | Total Samples ($k$) | | | | | |
| --- | --- | --- | --- | --- | --- | --- | --- |
| | | 8 | 16 | 32 | 64 | 128 | 512 |
| Amazon Polarity | KD | 0.628 ±0.055 | 0.690 ±0.034 | 0.766 ±0.032 | 0.827 ±0.021 | **0.835** ±0.037 | 0.846 ±0.009 |
| | + CoD | **0.697** ±0.117 | **0.782** ±0.033 | **0.823** ±0.018 | **0.844** ±0.009 | 0.814 ±0.013 | **0.855** ±0.018 |
| | LWD | 0.660 ±0.061 | 0.699 ±0.044 | 0.777 ±0.042 | 0.825 ±0.015 | **0.839** ±0.015 | 0.839 ±0.013 |
| | + CoD | **0.712** ±0.039 | **0.743** ±0.051 | **0.811** ±0.016 | **0.832** ±0.015 | 0.830 ±0.010 | **0.850** ±0.013 |
| CoLA | KD | 0.724 ±0.045 | 0.735 ±0.052 | 0.776 ±0.026 | 0.773 ±0.040 | 0.799 ±0.011 | 0.806 ±0.004 |
| | + CoD | **0.752** ±0.042 | **0.766** ±0.018 | **0.790** ±0.012 | **0.799** ±0.004 | **0.803** ±0.008 | **0.817** ±0.007 |
| | LWD | **0.699** ±0.042 | 0.744 ±0.039 | 0.755 ±0.043 | 0.787 ±0.008 | **0.803** ±0.009 | 0.808 ±0.008 |
| | + CoD | 0.685 ±0.190 | **0.780** ±0.018 | **0.790** ±0.004 | **0.798** ±0.007 | 0.802 ±0.005 | **0.813** ±0.003 |
| IMDB | KD | 0.743 ±0.070 | 0.849 ±0.037 | 0.882 ±0.032 | **0.904** ±0.004 | **0.912** ±0.005 | **0.920** ±0.004 |
| | + CoD | **0.893** ±0.007 | **0.896** ±0.007 | **0.900** ±0.005 | 0.904 ±0.005 | 0.910 ±0.008 | 0.918 ±0.003 |
| | LWD | 0.773 ±0.034 | 0.823 ±0.041 | 0.876 ±0.027 | **0.903** ±0.008 | **0.915** ±0.007 | 0.914 ±0.014 |
| | + CoD | **0.877** ±0.022 | **0.888** ±0.006 | **0.900** ±0.005 | 0.902 ±0.009 | 0.911 ±0.008 | **0.921** ±0.001 |
| SST2 | KD | 0.591 ±0.040 | 0.666 ±0.030 | 0.754 ±0.047 | 0.816 ±0.024 | 0.861 ±0.015 | 0.887 ±0.033 |
| | + CoD | **0.685** ±0.112 | **0.763** ±0.084 | **0.829** ±0.028 | **0.850** ±0.015 | **0.862** ±0.016 | **0.905** ±0.011 |
| | LWD | 0.580 ±0.064 | 0.664 ±0.024 | 0.726 ±0.036 | 0.818 ±0.019 | 0.847 ±0.029 | **0.912** ±0.005 |
| | + CoD | **0.658** ±0.107 | **0.675** ±0.074 | **0.839** ±0.017 | **0.841** ±0.019 | **0.859** ±0.016 | 0.877 ±0.044 |
| Yelp | KD | 0.704 ±0.062 | **0.793** ±0.042 | 0.861 ±0.011 | 0.887 ±0.004 | **0.907** ±0.007 | **0.922** ±0.008 |
| | + CoD | **0.759** ±0.086 | 0.758 ±0.084 | **0.870** ±0.008 | **0.889** ±0.009 | 0.897 ±0.009 | 0.920 ±0.006 |
| | LWD | 0.714 ±0.049 | **0.815** ±0.028 | 0.870 ±0.013 | 0.875 ±0.012 | **0.907** ±0.006 | **0.925** ±0.006 |
| | + CoD | **0.758** ±0.069 | 0.757 ±0.082 | **0.873** ±0.012 | **0.884** ±0.007 | 0.894 ±0.009 | 0.919 ±0.006 |
| Sent140 | KD | **0.580** ±0.032 | 0.594 ±0.026 | 0.634 ±0.047 | 0.681 ±0.046 | 0.740 ±0.012 | **0.796** ±0.013 |
| | + CoD | 0.573 ±0.078 | **0.612** ±0.064 | **0.721** ±0.019 | **0.737** ±0.030 | **0.767** ±0.014 | 0.795 ±0.006 |
| | LWD | **0.576** ±0.038 | 0.585 ±0.025 | 0.624 ±0.029 | 0.684 ±0.044 | 0.728 ±0.035 | **0.799** ±0.007 |
| | + CoD | 0.561 ±0.064 | **0.592** ±0.050 | **0.681** ±0.043 | **0.723** ±0.025 | **0.763** ±0.019 | 0.773 ±0.026 |

vs. 68.5%). These gains demonstrate the generality of our approach: it is effective even for decoder transformer families like `Qwen2.5`.

Taken together, our findings affirm the broad applicability of CFE-infused distillation. The consistent improvements across datasets, model families, and student capacities support our central claim: CFEs are a powerful, data-efficient tool for improving teacher-student alignment in low-resource scenarios.

### C.6 Ablation Results

We perform the following ablations: (1) *On template designing and prompt choices*. We vary prompt templates for generating CFEs and observed that CoD is robust to prompt choices, showing low standard deviation across variants and consistently outperforming the KD baseline few shot settings (see Table 7). One possible direction could be to use automatic prompt generation methods [72], however these are typically more compute-intensive. (2) *Computational and Memory Requirements*. We assess the computational efficiency of CoD under varying few-shot budgets $k$ using the `codecarbon` package [73] to track runtime and energy consumption (see Table 8). (3) *Effect of soft-label supervision*. To study the role of soft-labels in our approach, we remove or corrupting the teacher's soft labels (see Table 9). Removing the soft label term ($\alpha = 0$) leads to a substantial drop in performance across all shot levels. Although CoD still improves over KD in this setting, the gains are significantly reduced. This highlights that the soft label calibration from the teacher is a key contributor to the effectiveness of counterfactual explanation data. Additionally, when replacing soft labels with random values, performance degrades sharply, likely due to inconsistency with the hard labels, introducing conflicting supervision signals in the training objective.

Table 6: **Classification accuracy ($\pm$ std) of `Qwen2.5` across datasets with varying training sizes $k$.** For CoD, training data consists of $k/2$ standard and $k/2$ CFEs. Teacher model is `Qwen2.5-1.5B` and student model is `Qwen2.5-0.5B`.

| Dataset | Method | Total Samples ($k$) | | | | | |
|---|---|---|---|---|---|---|---|
| | | 8 | 16 | 32 | 64 | 128 | 512 |
| CoLA | KD | 0.681 ±0.012 | 0.676 ±0.023 | 0.668 ±0.042 | 0.654 ±0.032 | 0.676 ±0.020 | 0.732 ±0.014 |
| | + CoD | **0.683** ±0.016 | **0.686** ±0.018 | **0.697** ±0.015 | **0.711** ±0.020 | **0.736** ±0.017 | **0.757** ±0.011 |
| | LWD | 0.681 ±0.012 | 0.657 ±0.031 | 0.678 ±0.018 | 0.650 ±0.039 | 0.636 ±0.029 | 0.712 ±0.014 |
| | + CoD | **0.682** ±0.018 | **0.687** ±0.013 | **0.704** ±0.010 | **0.714** ±0.020 | **0.719** ±0.022 | **0.755** ±0.013 |
| Yelp | KD | 0.684 ±0.021 | 0.759 ±0.040 | 0.827 ±0.030 | 0.861 ±0.017 | **0.887** ±0.012 | **0.920** ±0.010 |
| | + CoD | **0.745** ±0.029 | **0.779** ±0.048 | **0.828** ±0.072 | **0.886** ±0.007 | 0.883 ±0.010 | 0.916 ±0.008 |
| | LWD | 0.685 ±0.019 | 0.777 ±0.036 | 0.837 ±0.027 | 0.876 ±0.020 | **0.898** ±0.008 | **0.920** ±0.005 |
| | + CoD | **0.746** ±0.028 | **0.778** ±0.035 | **0.847** ±0.020 | **0.876** ±0.014 | 0.883 ±0.010 | 0.909 ±0.009 |
| Amazon Polarity | KD | 0.589 ±0.057 | 0.635 ±0.044 | 0.706 ±0.083 | 0.781 ±0.033 | **0.807** ±0.031 | **0.862** ±0.013 |
| | + CoD | **0.605** ±0.051 | **0.660** ±0.042 | **0.712** ±0.077 | **0.793** ±0.030 | 0.805 ±0.041 | 0.835 ±0.021 |
| | LWD | 0.589 ±0.057 | 0.628 ±0.096 | 0.680 ±0.052 | 0.779 ±0.026 | 0.823 ±0.027 | **0.858** ±0.015 |
| | + CoD | **0.607** ±0.051 | **0.662** ±0.060 | **0.692** ±0.080 | **0.795** ±0.041 | **0.823** ±0.023 | 0.853 ±0.020 |
| IMDB | KD | 0.678 ±0.054 | 0.758 ±0.079 | 0.817 ±0.057 | **0.890** ±0.017 | 0.903 ±0.012 | **0.926** ±0.003 |
| | + CoD | **0.800** ±0.054 | **0.845** ±0.061 | **0.877** ±0.038 | 0.889 ±0.014 | **0.912** ±0.010 | 0.921 ±0.003 |
| | LWD | 0.678 ±0.054 | 0.740 ±0.076 | 0.832 ±0.035 | 0.883 ±0.014 | 0.906 ±0.014 | **0.925** ±0.003 |
| | + CoD | **0.800** ±0.055 | **0.835** ±0.048 | **0.869** ±0.012 | **0.893** ±0.013 | **0.909** ±0.008 | 0.920 ±0.007 |
| SST2 | KD | 0.568 ±0.061 | 0.621 ±0.084 | 0.719 ±0.102 | **0.827** ±0.038 | **0.878** ±0.020 | **0.904** ±0.010 |
| | + CoD | **0.578** ±0.064 | **0.663** ±0.081 | **0.767** ±0.085 | 0.779 ±0.137 | 0.870 ±0.019 | 0.886 ±0.005 |
| | LWD | 0.568 ±0.062 | 0.642 ±0.107 | 0.704 ±0.065 | **0.825** ±0.034 | **0.869** ±0.026 | **0.890** ±0.010 |
| | + CoD | **0.577** ±0.063 | **0.677** ±0.076 | **0.782** ±0.133 | 0.779 ±0.085 | 0.792 ±0.118 | 0.878 ±0.011 |
| Sent140 | KD | **0.586** ±0.047 | **0.599** ±0.047 | **0.641** ±0.030 | 0.708 ±0.027 | 0.756 ±0.020 | **0.813** ±0.010 |
| | + CoD | 0.556 ±0.038 | 0.591 ±0.046 | 0.616 ±0.055 | **0.711** ±0.061 | **0.757** ±0.023 | 0.805 ±0.010 |
| | LWD | **0.587** ±0.051 | **0.596** ±0.038 | **0.639** ±0.063 | **0.718** ±0.038 | 0.765 ±0.024 | 0.805 ±0.011 |
| | + CoD | 0.556 ±0.038 | 0.588 ±0.059 | 0.621 ±0.051 | 0.715 ±0.059 | **0.765** ±0.012 | **0.805** ±0.008 |

Table 7: **Comparison between KD and CoD variants across varying prompt templates**. CoD is robust to prompt choices, showing low standard deviation across variants and consistently outperforming the KD baseline in few shot settings.

| Method | 8 | 16 | 32 | 64 | 128 | 512 |
|---|---|---|---|---|---|---|
| KD | 0.617 | 0.712 | 0.757 | 0.820 | 0.848 | 0.899 |
| + CoD (v1) | 0.719 | 0.781 | 0.821 | 0.827 | 0.853 | 0.892 |
| + CoD (v2) | 0.754 | 0.789 | 0.841 | 0.872 | 0.890 | 0.872 |
| + CoD (v3) | 0.738 | 0.778 | 0.819 | 0.835 | 0.856 | 0.901 |
| + CoD (v4) | 0.734 | 0.783 | 0.830 | 0.834 | 0.883 | 0.888 |
| CoD *(mean)* | 0.736 | 0.783 | 0.828 | 0.842 | 0.870 | 0.888 |
| *(std)* | 0.012 | 0.004 | 0.009 | 0.018 | 0.016 | 0.010 |

Table 8: **Compute and energy use on SST2 for KD+CoD and LWD+CoD.** Accuracy, runtime, and energy increases with $k$. LWD+CoD is consistently costlier due to intermediate representation alignment.

| Method | $k$ | Accuracy | Duration (s) | CPU (kWh) | GPU (kWh) | RAM (kWh) |
|---|---|---|---|---|---|---|
| KD + CoD | 8 | 0.719 | 478.13 | 0.01336 | 0.00966 | 0.02479 |
| | 16 | 0.781 | 488.04 | 0.01341 | 0.00972 | 0.02499 |
| | 32 | 0.821 | 491.14 | 0.01382 | 0.01041 | 0.02547 |
| | 64 | 0.827 | 547.58 | 0.01472 | 0.13620 | 0.02723 |
| | 128 | 0.853 | 569.50 | 0.01639 | 0.01634 | 0.02952 |
| | 512 | 0.892 | 705.21 | 0.03120 | 0.03593 | 0.04536 |
| LWD + CoD | 8 | 0.694 | 485.12 | 0.01263 | 0.01102 | 0.02514 |
| | 16 | 0.785 | 496.07 | 0.01394 | 0.01158 | 0.02572 |
| | 32 | 0.832 | 517.78 | 0.01472 | 0.01245 | 0.02654 |
| | 64 | 0.830 | 536.01 | 0.01515 | 0.01311 | 0.02775 |
| | 128 | 0.835 | 668.52 | 0.01882 | 0.01670 | 0.02960 |
| | 512 | 0.880 | 814.65 | 0.04621 | 0.04712 | 0.04902 |

Table 9: **Effect of soft-label calibration on downstream performance**. Removing the soft label term ($\alpha = 0$) leads to a substantial drop in performance across all shot levels. Although CoD still improves over KD in this setting, the gains are significantly reduced. This highlights that the soft label calibration from the teacher is a key contributor to the effectiveness of counterfactual explanation data. Additionally, when replacing soft labels with random values, performance degrades sharply, likely due to inconsistency with the hard labels, introducing conflicting supervision signals in the training objective.

| Method (SST2) | 8 | 16 | 32 | 64 | 128 | 512 |
|---|---|---|---|---|---|---|
| KD (no soft label, $\alpha$=0) | 0.553 | 0.622 | 0.697 | 0.712 | 0.791 | 0.815 |
| + CoD (no soft label, $\alpha$=0) | 0.613 | 0.651 | 0.701 | 0.727 | 0.793 | 0.792 |
| KD (random soft label) | 0.582 | 0.533 | 0.543 | 0.601 | 0.617 | 0.649 |
| + CoD (random soft label) | 0.573 | 0.548 | 0.552 | 0.602 | 0.623 | 0.632 |
| KD (default) | 0.617 | 0.712 | 0.757 | 0.820 | 0.848 | 0.899 |
| + CoD (default) | 0.719 | 0.781 | 0.821 | 0.827 | 0.853 | 0.892 |

