# OpenReview forum: "Few-Shot Knowledge Distillation for Language Models via Counterfactual Explanations"
_NeurIPS.cc/2025/Workshop/Reliable_ML — NeurIPS 2025 - Reliable ML Workshop_

### Official Review · Reviewer_kayP · 2025-09-17
**This paper proposes a method to train a student model with a teacher model by additionally infusing counterfactual explanations**

**Rating:** 7
**Confidence:** 3

**Review:**

Summary: This paper discusses an interesting topic. Using counterfactual explanation to get a precise decision boundary in data data-poor regime.

Strengths:

1. Provides a theoretical analysis for a toy example.

2. A considerable number of empirical studies with enough implementation details.

3. Geometric perspective for better understanding

Weaknesses:

1. It seems to be only applicable as a classification model for LLMs rather than a generative model. How can we extend these ideas to generative applications?

2. We use LLM for counterfactual data generation, which invalidates its use case in a data-poor regime.

3. For the toy example, we choose the counterexample as the one with the minimum perturbation to the original data. How can we enforce it on the LLM-generated counterfactual explanations?

However, I lean toward acceptance, since it tries to balance theory with application.

---

### Official Review · Reviewer_nBux · 2025-09-19
**Review for "Few-Shot Knowledge Distillation for Language Models via Counterfactual Explanations"**

**Rating:** 7
**Confidence:** 2

**Review:**

The paper studies teacher-student knowledge distillation in the few-shot regime and introduces a novel approach to the problem called Counterfactual Explanation-infused Distillation (COD). While knowledge distillation has proven effective for compressing large language models, its reliance on large datasets makes it less practical in data-scarce settings. By leveraging counterfactual explanations to enrich few-shot training sets, the authors aim to better capture the teacher model’s decision boundaries with fewer examples. Thus, I find the problem very well-motivated.

Strengths:

- The paper bridges explainability and model compression by turning counterfactuals into actionable training signals. The combination of theoretical guarantees and empirical results strengthens the case for the approach, and the experiments cover a good range of datasets.

- The paper provides both theoretical and empirical results that are quite solid and seem to back up the intuition that COD is the correct approach in the few-shot regime.

Weaknesses:
- Generating counterfactuals can be computationally expensive, and the method relies heavily on the quality of the teacher model, meaning that biases or errors could be transferred.

- The training pipeline is more complex, both in terms of generating counterfactuals and ensuring their quality. This could make adoption less straightforward compared to simpler distillation techniques.

Overall, I liked the paper and this that it makes solid and contributions on a well-motivated problem relevant to ongoing discussions of the ML community on data efficiency. I feel that the approach could be impactful, particularly in real-world cases where labeled data is scarce but efficient models are needed.